



# Flutter Behavior of Highly Flexible Two- and Three-bladed Wind Turbine Rotors

Mayank Chetan[1], Shulong Yao[1], and D. Todd Griffith[1]

[1]UTD Center for Wind Energy, Department of Mechanical Engineering, The University of Texas at Dallas, Richardson, Texas, USA

**Correspondence:** D. Todd Griffith (tgriffith@utdallas.edu)

**Abstract.** With the progression of novel design, material, and manufacturing technologies, the wind energy industry has successfully produced larger and larger wind turbine rotor blades while driving down the Levelized Cost of Energy (LCOE). Though the benefits of larger turbine blades are appealing, larger blades are prone to aero-elastic instabilities due to their long, slender, highly flexible nature, and this effect is accentuated as rotors further grow in size. In addition to the trend of larger rotors, new rotor concepts are emerging including two-bladed rotors and downwind configurations. In this work, we introduce a comprehensive evaluation of flutter behavior including classical flutter, edgewise vibration, and flutter mode characteristics for two-bladed, downwind rotors. Flutter speed trends and characteristics for a series of both two- and three-bladed rotors are analyzed and compared in order to illustrate the flutter behavior of two-bladed rotors relative to more well-known flutter characteristics of three-bladed rotors. In addition, we examine the important problem of blade design to mitigate flutter and present a solution to mitigate flutter in the structural design process. A study is carried out evaluating the effect of leading edge and trailing edge reinforcement on flutter speed and hence demonstrates the ability to increase the flutter speed and satisfy structural design requirements (such as fatigue) while maintaining or even reducing blade mass.

## 1 Introduction

With increase in demand and reduction in costs, wind energy offers a promising clean energy solution. The wind industry has successfully produced large-scale wind turbines by relying on new materials, design, and manufacturing technologies while driving down the Levelized Cost of Energy (LCOE). However, with the increase in rotor size, aero-elastic instabilities like flutter become a significant concern and should be actively examined in the design process. In addition to rotor size impacts on flutter behavior, various novel turbine concepts have been studied as a possible pathway to enable turbine designs at these extreme scales. One such concept is the use of two-bladed, downwind turbines (Ichter et al., 2016) at extreme-scales in a project called SUMR (Segmented Ultralight Morphing Rotor). The various studies within the SUMR project have shown that two-bladed downwind turbines can provide a significant reduction in LCOE (Loth et al., 2017; Yao et al., 2021b, a; Zalkind et al., 2019; Pao et al., 2021; Kaminski et al., 2020a) as well as allow for the design of large turbines up to 50MW (Yao et al., 2021b; Martin, 2019). The flutter behavior of these two-bladed designs are interesting and important to examine due to both



their highly flexible nature and structural differences for two-bladed versus three-bladed rotors (Yao et al., 2021a), which is the focus of the present study.

A number of studies have been performed over recent decades investigating aero-elastic stability, including classical flutter, in horizontal axis wind turbine rotors, but all focused on the conventional upwind, three-bladed rotor configuration. Riziotis and Madsen (Riziotis and Madsen, 2011) define classical flutter as the instability occurring from the aeroelastic coupling of the flapwise modes with the torsion modes during operation. In particular, the change in the angles of attack resulting from the torsion deformation of the wing sections generates aerodynamic lift forces that are in phase with the flapwise bending motion. This gives rise to violently amplifying flapwise vibrations that cannot be compensated by structural damping. Lobitz (Lobitz, 2004) approached the problem of flutter in MW-sized wind turbines with a NASTRAN-based beam finite element model with Theodorsen unsteady aerodynamics (Theodorsen, 1934) for a blade rotating in still air, a common assumption in flutter analysis where the inflow component of wind velocity is neglected and only the rotor plane blade velocity is included. Pourazarm et al. (Pourazarm et al., 2016) developed and validated a flutter model against previous work, using a blade model that accounts for flapwise and torsional degrees of freedom, although the edgewise degree of freedom was neglected in their formulation. Owens et al. (Owens et al., 2013) developed the BLAST tool for flutter prediction, which was similar to the approach of Lobitz, but with an improved structural model including flapwise, edgewise, and torsional degrees of freedom. Modifications were made to the mass, stiffness, and damping matrices to account for aerodynamic effects, and rotational effects such as Coriolis effects and spin softening. Hansen et al. (Hansen, 2004) further developed a full turbine model in still air using an eigenvalue approach based on a beam finite element (FE) model with aerodynamic loads modeled using the blade element momentum (BEM) theory coupled with the Leishman-Beddoes Dynamic stall model. They concluded that there was a reasonable similarity between flutter predictions for a full turbine versus analysis of an isolated blade uncoupled from the tower. More recently, Farsadi et al. (Farsadi and Kayran, 2021) developed a method to account for compressibility effects on the wind turbine blades, but the resulting flutter speeds were comparable to the classical flutter analysis methods developed by Hansen (Hansen, 2007).

In addition to classical flutter involving coupled flapwise and torsional modes, edgewise vibration has also been shown to be a concern for large wind turbine rotor blades. In the work of Griffith and Chetan (Griffith and Chetan, 2018), it was shown how larger blades tend to have a larger edgewise contribution to blade instability as blade structural designs at the 100-meter scale are optimized for mass. These edgewise instabilities in large turbine blades have been observed experimentally and numerically by Kallesøe (Kallesøe and Kragh, 2016). The edgewise flutter-like instabilities were found to have a shallow crossover that is considered as "soft" crossover to an unstable mode due to higher structural damping. Additionally, the edgewise instabilities resulted in limit cycle oscillations and were confirmed by further experimentation on a Siemens 7MW turbine by Volk et al. (Volk et al., 2020). Bergami (Bergami, 2008) examined the addition of the edgewise unsteady aerodynamic terms and reached the conclusion that this addition did not have much impact on flutter predictions. Kelly et al. (Kelley and Paquette, 2020) made an improvement to the BLAST tool (Owens et al., 2013) also by including an edgewise term for the unsteady aerodynamics of the blade; however, the same conclusion was reached in that this addition did not have much impact on the predicted flutter speeds.





In the present study, we focus on the flutter performance of two- and three-bladed designs. Prior work focused on the analysis of various three-bladed wind turbine blade designs and understanding their performance. A study on flutter instabilities of two-

bladed wind turbine blade designs has not been studied, and here we introduce and present an examination of flutter behavior for two-bladed rotors. Here, we investigate the sensitivity and trends of flutter predictions, including the flutter speeds and flutter mode shapes, for a series of conventional three-bladed upwind rotors and a series of downwind two-bladed rotors. Critical comparisons are made that demonstrate clear differences in trends and characteristics of flutter modes for two- versus three-bladed wind turbine rotors. Once the flutter behavior is determined, it is important to examine blade designs that mitigate

flutter instability. Previous works (Griffith and Chetan, 2018; Chetan et al., 2019a, b) have carried out blade design studies on the composite layups of the wind turbine blades to achieve a higher flutter margin. Thus, in the final analysis, a detailed trend study is conducted to evaluate the effect of trailing edge and leading edge reinforcements on the flutter margins of two- and three-bladed turbines using a semi-automated design tool called AutoNuMAD.

The paper is organized as follows: Section 2 presents the methodology for flutter prediction implemented in this paper. In

Section 3, the various wind turbine blade models evaluated in this study are presented along with the bench-marking of the flutter evaluation tool. In Section 4, the flutter behavior for the various two- and three-bladed wind turbine blades are analyzed. Section 5 presents the design space exploration for two- and three-bladed wind turbine blades to improve their flutter margins. In Section 6, concluding remarks and future research directions are summarized.

## 2   Method for Classical Flutter Prediction

In this section, we present the methods utilized for flutter prediction used in this study. First, we consider the incorporation of the rotational effects into the model. The system of equations for the blade including rotational effects such as Coriolis and spin softening takes the form of Eq. 1. This equation is based on the 3-D implementation of the Euler-Bernoulli beam finite element, thus allowing for the blade to deform in flapwise, edgewise, and torsional components represented by $x$.

$$M\ddot{x} + (D + G(\Omega))\dot{x} + (K(x) - S(\Omega))x = F_{cent}(\Omega) + F \qquad (1)$$

Here $M$, $D$, $K(x)$ are the mass, damping, and stiffness matrices of the blade structure, respectively, that result from the Euler-Bernoulli finite element formulation. Here, the stiffness matrix $K(x)$ is a function of the geometry to account for potential geometric non-linearities. For the rotational effects, $G(\Omega)$ introduces the Coriolis matrix and $S(\Omega)$ represents the spin softening effects at an angular velocity of $\Omega$. $F$ represents the non-potential forces, and $F_{cent}(\Omega)$ is the centrifugal forces on the system.

Now we consider aerodynamic effects in the flutter model. For aerodynamic effects, the Theodorsen unsteady aerodynamic

theory (Theodorsen, 1934) is introduced and the lift and moment equations that cause the flapping and twisting motion at the defined cross-sections are given by Eq 2 and Eq. 3 respectively.

$$L = \pi \rho b^2 \left[ \ddot{z} + V\dot{\theta} - ba\ddot{\theta} \right] + 2\pi \rho V b C(k) \left[ \dot{z} + V\theta + b(\frac{1}{2} - a)\dot{\theta} \right] \qquad (2)$$





$$M_p = \pi \rho b^2 \left[ ba\ddot{z} - Vb \left( \frac{1}{2} - a \right) \dot{\theta} - b^2 \left( \frac{1}{8} + a^2 \right) \ddot{\theta} \right] + 2\pi \rho V b^2 \left( a + \frac{1}{2} \right) C(k) \left[ \dot{z} + V\theta + b \left( \frac{1}{2} - a \right) \dot{\theta} \right] \tag{3}$$

Here $\rho$ is the air density, $b$ is the semi-chord of the airfoil section, $a$ is the elastic axis position aft of the mid chord as a
fraction of the semi-chord, $z(t)$ represents the flapwise motion for the sections, $\theta(t)$ is the torsional motion of the section and
$C(k)$ is the Theodorsen Function (Wendell, 1982) which models the amplitude and phase lag of the aerodynamic forces acting
on the section. Figure 1 illustrates a 2-D airfoil undergoing heave and pitch motion.

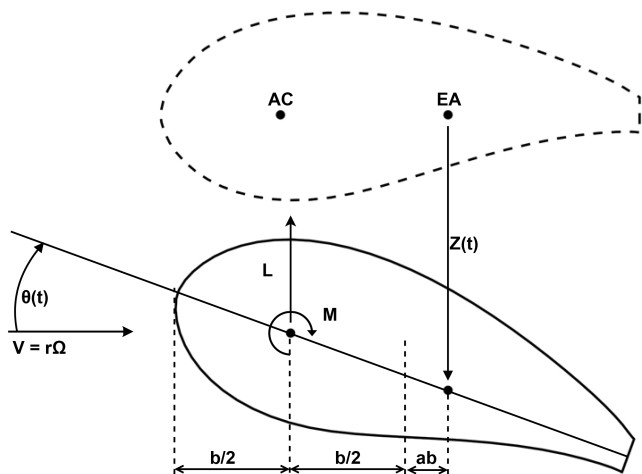

**Figure 1.** Illustration of a two-dimensional airfoil undergoing heave and pitch motion

$$C(k) = F_{th}(k) + iG_{th}(k) \tag{4}$$

$$k = \frac{\omega b}{U_\infty} \tag{5}$$

Here $k$ is the reduced frequency and depends on the oscillatory motion of the airfoil section. The freestream velocity, $U_\infty$ is
modeled for a rotating turbine as a function of distance from the hub axis $r$ given by:

$$U_\infty = r\Omega \tag{6}$$

The resulting aerodynamic loads are a function of the angular velocity $\Omega$ and the frequency $\omega$. Now, modeling the aerody-
namic mass, damping, and stiffness into Eq. 1 results in:

$$(M + M_A(\Omega))\ddot{x} + (D + G(\Omega) + D_A(\Omega, \omega))\dot{x} + (K(x) - S(\Omega) + K_A(\Omega, \omega))x = F_{cent}(\Omega) + F_A(\Omega) \tag{7}$$

Where, $M_A(\Omega), D_A(\Omega, \omega)$ & $K_A(\Omega, \omega)$ are the aerodynamic mass, damping and stiffness matrices, respectively. The vector
$F_A(\Omega)$ represents non-potential forces like aerodynamic forces. The coefficients of Equation 7 are dependent on the rotor





speed $\Omega$ and reduced frequency $k$. The geometric non-linearities can be linearized about the operating rotor speed by solving
the nonlinear static elasticity equation:

$$[K(x) - S(\Omega)]\,x = F_{cent}(\Omega) \tag{8}$$

After solving for the equilibrium configuration $x_{eq}$, the stiffness matrix can be updated to $K(x_{eq})$. Finally, considering the left-hand side for Equation 7 and substituting $K(x_{eq})$; we have the below equation on which the modal analysis can be carried out.

$$(M + M_A(\Omega))\ddot{x} + (D + G(\Omega) + D_A(\Omega,\omega))\dot{x} + (K(x_{eq}) - S(\Omega) + K_A(\Omega,\omega))x = 0 \tag{9}$$

Next, Equation 9 is solved using an iterative procedure called "p-f iteration" (Wright and Cooper, 2008), where the initial frequency of the system is guessed and iterated over until convergence. The modes of the system are analyzed for negative damping which indicates potential aeroelastic instability for that particular mode. This procedure is carried out iteratively for the entire rotational speed range of interest.

This method and similar methods have been used reliably in the past for flutter predictions (Owens et al., 2013; Lobitz, 2004). However, we now consider the assumptions of this model. Namely, the main assumptions are: (1) the flow is always attached to the airfoil section, (2) the airfoil is thin, (3) the resulting wake is also flat (parallel to the rotor plane inflow), (4) the blade rotates in still air, and (5) the wake of the blade does not affect the nearing blade. Assumptions 1, 2, and 3 are assumptions of the Theodorsen theory, which have been found to be reasonable assumptions over the majority of the blade span (Hansen,
2004, 2007; Owens et al., 2013; Lobitz, 2004; Kelley and Paquette, 2020).

The still air assumption (Assumption 4) is an important assumption for the flutter prediction procedure of wind turbine blades and is now considered. Of course, the wind velocity at each span-wise section of the blade depends upon the inflow component to the rotor plus the local in-plane rotor plane component (due to $r\Omega$). The still air assumption considers only the $r\Omega$ component. One way to evaluate the impact of the inflow velocity effect on flutter speed is to directly include it in the model,
and this has been performed by Hafeez et al. (Abdel Hafeez and El-Badawy, 2018), where the Theodorsen function (Wendell, 1982) was modified to account for inflow along with the rotor plane velocity. They show that including the inflow resulted in about a 5% increase in flutter speed versus the still air assumption. The conclusions are that the inflow has a small effect on flutter speed and that the still air assumption tends to be conservative (producing a lower flutter speed estimate), thus, the still air assumption is a reasonable and acceptable assumption.

**3    Description of Two- and Three-bladed Wind Turbines**

In order to examine the flutter behavior for large wind turbines, a number of wind turbine blade models are examined including open source, reference models for three-bladed rotors, and recently designed two-bladed rotors. Figure 2 shows an illustration of the various wind turbine models considered in this study.





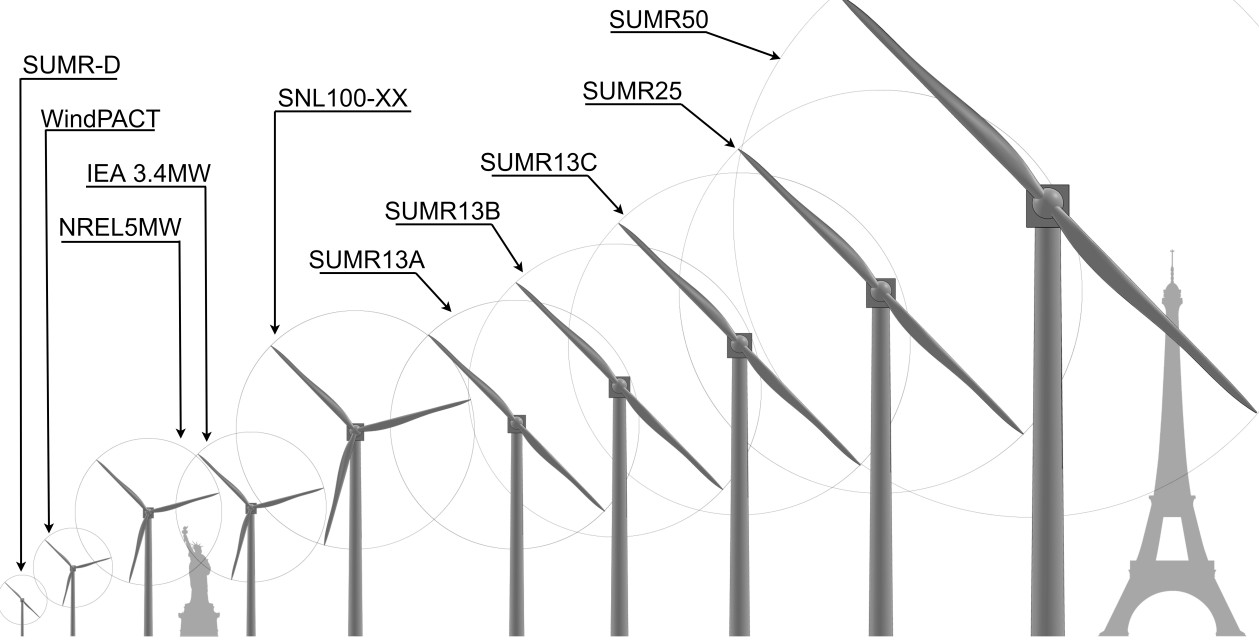

**Figure 2.** Illustration comparing the wind turbine models considered in this study of flutter behavior in two- and three-bladed rotors.

### 3.1 Three-bladed wind turbine models

For the analysis of flutter for three-bladed rotors, the following designs are analyzed: (1) the IEA 3.4MW Onshore Reference Wind Turbine (RWT) is a class IIIA land-based wind turbine (Bortolotti et al., 2019); (2) the NREL 5MW (Jonkman et al., 2009), which is an offshore wind turbine with 61.5m blade that was designed to be used as a baseline representing utility-grade offshore wind turbines; (3) the IEA 10MW Offshore RWT is a class IA offshore wind turbine with a rated power of 10MW, a direct-drive system, and a monopile foundation (Bortolotti et al., 2019); (4) the SNL100-00 13.2-MW (Griffith and Resor,

2011), a 100-meter blade designed at Sandia National Laboratories as a baseline for large wind turbine blade studies; (5) the SNL100-01 13.2-MW (Griffith, 2013a), an update to the SNL100-00 blade with carbon spar-cap; (6) the SNL100-02 13.2-MW (Griffith, 2013b), a lighter blade from the SNL100-01 through the use of advanced core materials; (7) the SNL100-03 13.2-MW (Griffith and Richards, 2014), the fourth and final design in the SNL100 series, involved a significant change in geometry and materials to achieve further mass reduction. In this design, flatback airfoils were incorporated instead of sharp

trailing edge airfoils, and a completely new aerodynamic design was developed; and (8) the UTD100-04 13.2-MW (Griffith and Chetan, 2018) a 100-meter blade that was developed based on the geometry of SNL100-03. This final design shows not only an improvement in flutter margin from SNL100-03 but is also a lighter blade. A summary of the properties of the blades are represented in Table 1.



**Table 1.** Geometry and operating specification of the three-bladed wind turbine blade models examined in the study. The blades are sorted by blade length.

| Blade | Length (m) | Maximum Chord | Mass (kg) | Rated Power | Rated Speed |
|---|---|---|---|---|---|
| WindPACT 1.5MW | 33.25 | 2.8 | 4,326 | 1.5 | 20.46 |
| NREL 5MW | 61.5 | 4.6 | 17,740 | 5 | 12.1 |
| IEA 3.4MW | 63 | 4.29 | 16,441 | 3.4 | 11.75 |
| SNL100-00 | 100 | 7.63 | 114,172 | 13.2 | 7.44 |
| SNL100-01 | 100 | 7.63 | 73,995 | 13.2 | 7.44 |
| SNL100-02 | 100 | 7.63 | 59,047 | 13.2 | 7.44 |
| SNL100-03 (CONR) | 100 | 5.22 | 49,519 | 13.2 | 7.44 |
| UTD100-04 | 100 | 5.22 | 49,126 | 13.2 | 7.44 |

## 3.2 Two-bladed wind turbine models

We now describe recently designed two-bladed rotors to be examined in this flutter prediction study: (1) the SUMR13A (Yao et al., 2021a; Zalkind et al., 2019; Pao et al., 2021), a 104.34-meter blade, is the initial blade in the SUMR13 series of blade designs aimed towards the goal of attaining a 25% mass reduction. It is a flatback airfoil design with carbon fiber spar-caps; (2) the SUMR13B (Yao et al., 2021a; Zalkind et al., 2019; Pao et al., 2021) is a 122.86-meter intermediate design with a slender aerodynamic design for a lower axial induction factor; (3) the SUMR13C (Yao et al., 2021a; Zalkind et al., 2019; Pao et al.,

2021), the final iteration in the SUMR13 blade series, is a 143.45-meter blade; (4) the SUMR-D (Yao et al., 2019; Bay et al., 2019; Kaminski et al., 2020b; Kaminski, 2020) blade is a 20.87-meter 1/5th scale subscale model of the SUMR13A design and currently under testing on the CART 2 (Fingersh and Johnson, 2002; Bossanyi et al., 2010) platform at the National Wind Technology Center; (5) the SUMR25 design with a rated power of 25MW and blade length of 169m (Qin et al., 2020), and (6) the SUMR50 (Yao et al., 2021b) design with a rated power of 50MW and blade length of 246.77m. A summary of the

properties of the blades are represented in Table 2.

**Table 2.** Geometry and operating specification of the two-bladed wind turbine blade models examined in the study. The blades are sorted by blade length.

| Blade | Length (m) | Maximum Chord | Mass (kg) | Rated Power | Rated Speed |
|---|---|---|---|---|---|
| SUMR-D | 20.87 | 1.56 | 847 | 0.39 | 21.96 |
| SUMR13A | 104.34 | 7.51 | 57,621 | 13.2 | 9.55 |
| SUMR13B | 122.86 | 6.79 | 102,170 | 13.2 | 7.99 |
| SUMR13C | 143.45 | 9.28 | 118,110 | 13.2 | 7 |
| SUMR25 | 169 | 10.78 | 142,100 | 25 | 5.9 |
| SUMR50 | 246.77 | 15.8 | 392,000 | 50 | 3.95 |





### 3.3 Flutter Tool Benchmarking

In this section, flutter speeds for a few of the blade models discussed in the previous section are analyzed and compared with flutter predictions from previously published studies. To gain a more thorough understanding, both critical flutter speeds and flutter mode shapes are analyzed. We find generally good agreement in comparing our predictions (current study) with those

of other studies (as noted in Table 3). The WindPACT blade shows a flutter speed of 38.06rpm with a second flap and first torsional mode. This flutter speed is lower than values reported by Pourazarm (Pourazarm et al., 2016) and Owens who also report a third flap and first torsion mode coupling. The flutter speed for the NREL 5MW blade in the current study is 20.80 rpm, which is close to the speed predicted by Pourazarm et al. but slightly lower than Hansen. Although our tool predicts a different flutter mode (2nd flapwise mode coupled with 1st torsion) than reported by Pourazarm and Hansen (Hansen, 2007).

The difference in predictions may be attributed to the fact that the edgewise degree of freedom is not considered by Pourazarm et al. The flutter speeds for the SNL100-XX series of 100-meter blades (SNL100-00, -01, -02, -03) are in very good agreement with previous studies (Owens et al., 2013; Griffith, 2013a, b; Griffith and Richards, 2014) and are on the conservative side. Note that the maximum rpm for the SNL100-XX series is 7.44 rpm and the flutter ratio (i.e.; flutter speed divided by maximum rotor rpm) is calculated for each and placed in parentheses for the current study.

**Table 3.** The flutter speeds in units of rpm: Current study and previous studies for a few blade models

| Blade | Pourazarm et al. | Hansen | Owens et al. | Griffith et al. | Current Study |
|---|---|---|---|---|---|
| WindPACT | 45.45 | - | 43.4 | - | 38.06 (1.86) |
| NREL 5MW | 20.7 | 24 | - | - | 20.80 (1.71) |
| SNL100-00 | 16.91 | - | 13.05 | - | 14.11 (1.89) |
| SNL100-01 | - | - | - | 13.69 | 13.19 (1.77) |
| SNL100-02 | - | - | - | 12.72 | 12.24 (1.65) |
| SNL100-03 | - | - | - | 10.41 | 10.45 (1.40) |

## 4 Analysis of Flutter Behavior for Two- and Three-bladed Rotors

We start the analysis by examining the trends in flutter predictions for the three-bladed designs following the method described in Section 2. As noted in the introduction, there are two families of flutter instabilities that we examine in this work for large-scale, highly flexible wind turbine rotors: (1) classical flap-torsion coupled-mode flutter instability and (2) edgewise mode dominated flutter instability referred to as edgewise vibration. In this analysis, we are interested to examine the behavior of

both types of unstable modes, thus we search for and analyze both types of modes following the procedure introduced in Section 2.

### 4.1 Three-bladed wind turbines

In analyzing various blade cases for the three-bladed rotors, we observed an edgewise dominated instability occurring in many
cases before classical flap-torsional coupled mode flutter. This trend exists for most three-bladed designs. Further examination
of the damping vs rpm plots (Figure 3) shows that the transition to these edgewise dominated instabilities are "soft" or gradual
transitions to the unstable region. Figure 4 illustrates one such edgewise instability for the SNL100-03 blade. Similar edge-
wise instabilities have also been reported in previous studies and in full-scale wind turbine stability experiments (Kallesøe and
Kragh, 2016; Volk et al., 2020). As designers, we note this is a very serious concern because the edgewise (in-plane) aerody-
namic damping is relatively very low compared to flapwise aerodynamic (out of plane) damping. And the in-plane modes can
be excited by in-plane rotor loads including gravitational and inertial effects. Again, classical flap-torsion coupled-mode flutter
(or simply classical flutter) is defined as the unstable coupling of a torsional mode with a flapwise mode. Typically, these tend
to be the first torsional mode coupled with the second or third flapwise modes, Figure 5 illustrates the modal contributions of
a classical flutter mode for the SNL100-03 blade.

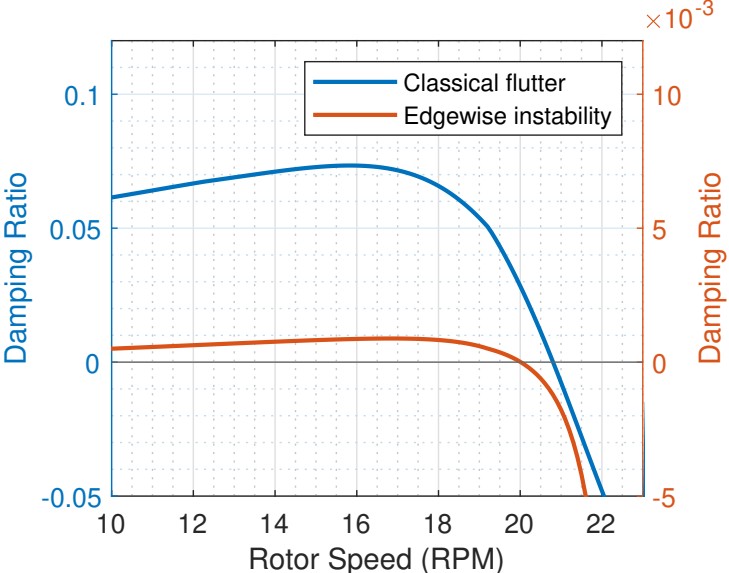

**Figure 3.** An illustration of a typical damping ratio vs. rotor speed plot. The Transition for edgewise instability and classical flutter is shown.
Please note the scale for the edgewise instability damping is $\frac{1}{10}^{th}$ that of classical flutter damping

Expanding on the nature of each three-bladed design, we start with the WindPACT 1.5MW turbine as presented in Table 4.
This blade follows the trend discussed above, where we have an edgewise instability occurring at 37.91rpm before classical
flutter which occurs at 38.06 rpm. Though the edgewise contribution in the first unstable mode is dominant it does have a small
degree of contribution from 2nd flapwise and 1st torsional modes. Next, the NREL 5MW design, which has a pure edgewise
unstable mode at 20.01 rpm and classical flutter occurring at 20.08 rpm with a flutter margin of 1.65. The NREL 5MW is a

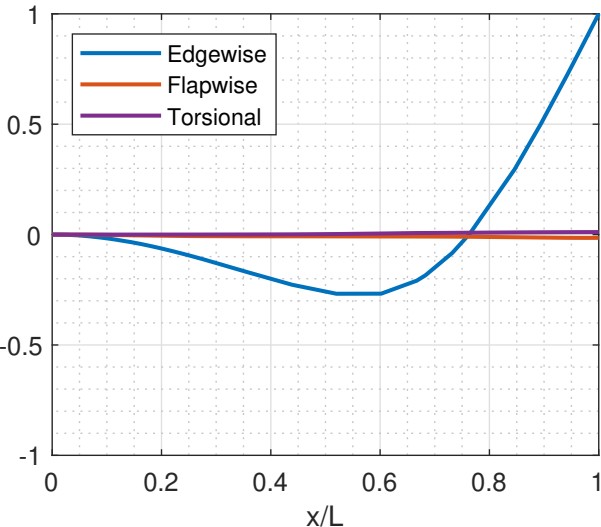

**Figure 4.** Illustration of 1st edgewise unstable mode for the SNL100-03 blade

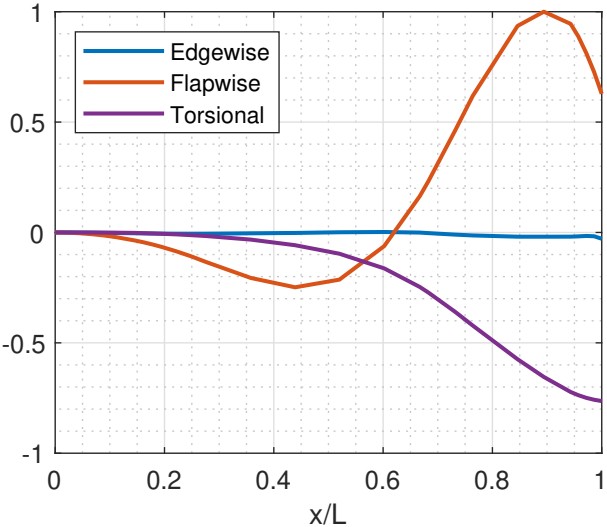

**Figure 5.** Illustration of 1st classical flutter mode for the SNL100-03 blade

traditionally designed blade representing the industry-standard offshore turbine in 2009. In contrast, for the IEA 3.4MW which
has a similar blade length as the NREL 5MW, the flutter margin is lower at 1.32. To note, the IEA 3.4MW does not have an
edgewise unstable mode occurring at a lower rpm than the unstable classical flutter mode as this blade shows and edgewise
instability at 33.8 rpm and classical flutter at 15.57 rpm. The IEA 3.4MW blade is an optimized blade with a lower rating,
this points to a trend of highly innovative, optimized blades tending to have a lower flutter margin. This behavior is further
examined for the SNL100-XX series in the following.



**Table 4.** Flutter speeds for three-bladed wind turbine models. E = Edgewise, F = Flapwise, T = Torsional. The numbers next to the mode type indicate the mode order. The modes are arranged according to highest to lowest contribution.

| Turbine | No of Blades | Blade Length | Rated RPM | 1st Unstable Edgewise mode | | | 1st Classical Flutter mode | | |
|---|---|---|---|---|---|---|---|---|---|
| | | | | RPM | Nature | Transition | RPM | Nature | Transition |
| WINDPact 1.5MW | 3 | 33.25 | 20.46 | 37.91 | E2; F2;T1 | Soft | 38.06 | F2;T1;E2 | Hard |
| NREL 5MW | 3 | 61.5 | 12.1 | 20.01 | E2 | Soft | 20.8 | F2;T1 | Hard |
| IEA 3.4MW | 3 | 65 | 11.75 | 33.8 | E1 | Soft | 15.57 | F3;T1 | Hard |
| SNL100-00 | 3 | 100 | 7.44 | 23.39 | E1 | Soft | 14.11 | F3;E2;T1 | Hard |
| SNL100-01 | 3 | 100 | 7.44 | 12.81 | E2 | Soft | 13.2 | F2;T1;E2 | Hard |
| SNL100-02 | 3 | 100 | 7.44 | 11.63 | E2 | Soft | 12.25 | F2;T1;E2 | Hard |
| SNL100-03 | 3 | 100 | 7.44 | 9.86 | E2 | Soft | 10.45 | F2;T1;E2 | Hard |
| UTD100-04 | 3 | 100 | 7.44 | 10.87 | E2 | Soft | 12.5 | F3;T1 | Hard |

Next, looking at the SNL100-XX blade series (SNL100-00, SNL100-01, SNL100-02, and SNL100-03) we note the trend to lower flutter speed (and lower flutter ratio) of the 100-meter designs as the mass of the designs was successively reduced in the blade optimization sequence. Note from Table 4 that for the SNL100-XX series the flutter speeds (rpm) are 14.11, 13.20, 12.25, and 10.45, respectively for SNL100-00 to SNL100-03. Except for the SNL100-00 blade, the SNL100-XX series of blades have a 2nd edgewise unstable mode before the occurrence of classical flutter. The trend in reduced flutter speed is clear for the first

three blades, which have the same airfoils and aerodynamic design, and this trend continues for the most lightweight SNL100-03 design, which was redesigned aerodynamically with a significantly smaller chord using flatback airfoils. In reviewing the span-wise stiffness properties, the trend to lower flapwise, edgewise, and torsional stiffnesses are evident (Figure 6). This is the main driver of reduced flutter speed in the SNL100-XX series. Finally, the UTD100-04 blade (Griffith and Chetan, 2018) which is a re-design of the SNL100-03 model exhibits similar behavior to the SNL100-03 blade but with a higher flutter speed

of 12.50rpm.

In examining the stiffness distributions of the SNL100-XX series of blades certain patterns emerge that can help explain the reductions in flutter speed. First, we can notice the significantly higher edgewise stiffness of the SNL100-00 blade. As discussed earlier the SNL100-00 blade is the only one in the SNL100-XX series that does not have an edgewise unstable mode before classical flutter. Next, looking at the flapwise and torsional stiffness distributions of the blades we can observe the

progressive reductions along the generations in this design series. This correlates directly to the decrease in the flutter speeds that were computed for the blades.

In addition to the stiffness distribution along the span, the chordwise location of the elastic axis (EA), mass center (CG), and the aerodynamic center (AC) of the blade contribute to the flutter performance of the blade. These results are included in the Appendix in Figures A1-A8 show the various axis for the three-bladed designs. Typically, in blade design, the elastic axis is

adjusted so that it aligns with the reference/pitch axis of the blade. Three-bladed wind turbine blades tend to have a lower chord



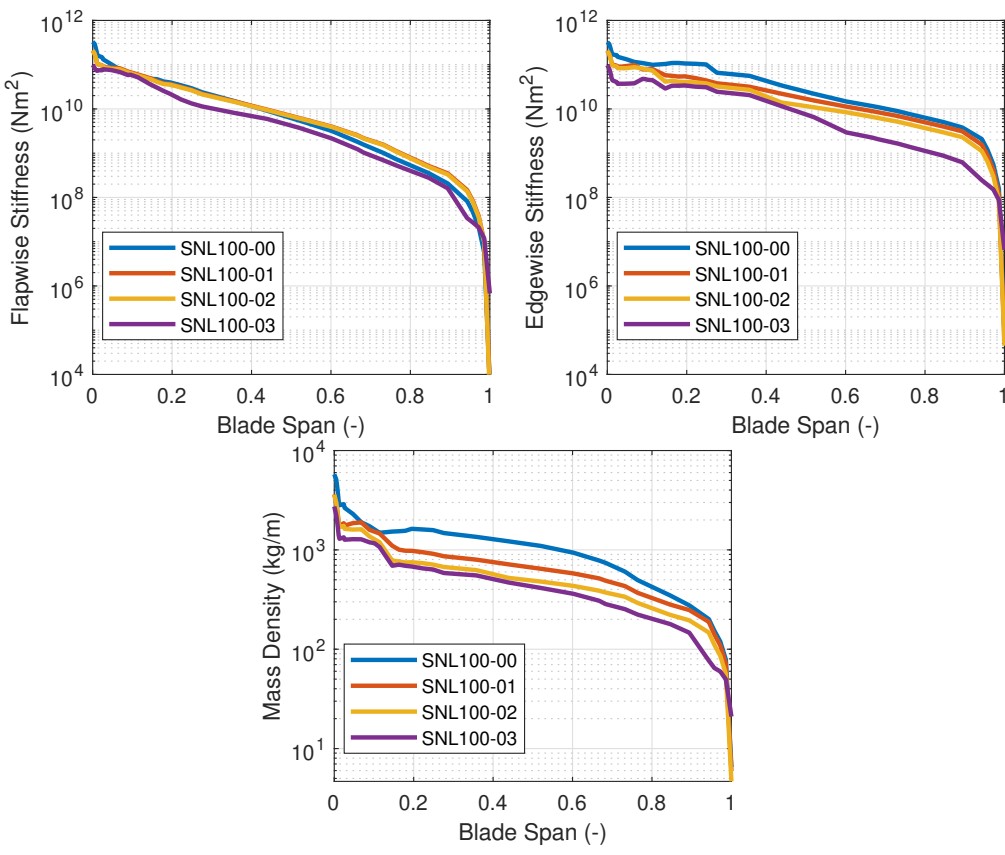

**Figure 6.** Flapwise stiffness, edgewise stiffness, and mass properties for the SNL100-XX series of blades

value in comparison to two-bladed designs to maintain the solidity of the turbine. This in turn contributes to lower stiffness for three-bladed designs as well as larger margins between the EA and CG locations in the chordwise direction.

## 4.2 Two-bladed wind turbines

We now turn our attention to two-bladed rotor designs. In looking at aero-structural optimized two-bladed wind turbine designs
(e.g.; SUMR13A or SUMR13C) we can observe that they tend to have much larger chords versus aero-structural optimized three-bladed rotors (e.g.; SNL100-03) as shown in Tables 1&2. This results from the solidity of the two- and three-bladed rotors being similar, thus requiring larger chords for two-bladed rotors. Table 5 shows a comparison of unstable edgewise and classical flutter modes for a series of two-bladed designs. In each case, the contributions of the different modes are described as well as the nature of transition to the unstable mode. From Table 5, we also observe that the flutter rpm's decrease with
the increase in the blade lengths. An interesting finding is that, unlike the three-bladed designs, the two-bladed designs do not exhibit an edgewise instability at an rpm lower than the rpm of the classical flutter mode. This is primarily due to two-bladed designs having higher edgewise stiffness than the three-bladed designs, which is largely an artifact of the larger chord.



**Table 5.** Flutter speeds for two-bladed wind turbine models. E = Edgewise, F = Flapwise, T = Torsional. The numbers next to the mode type indicate the mode order. The modes are arranged according to highest to lowest contribution.

| Turbine | No of Blades | Blade Length | Rated RPM | 1st Unstable Edgewise mode | | | 1st Classical Flutter mode | | |
|---|---|---|---|---|---|---|---|---|---|
| | | | | RPM | Nature | Transition | RPM | Nature | Transition |
| SUMR-D | 2 | 20.9 | 21.96 | 58.5 | E2;F3 | Soft | 53.98 | T1; F2 | Hard |
| SUMR13A | 2 | 104.35 | 9.55 | 16.2 | E2 | Soft | 10.4 | T1; F2;E2 | Hard |
| SUMR13B | 2 | 123 | 7.99 | 10.4 | E2 | Soft | 8.23 | F3; T1 | Hard |
| SUMR13C | 2 | 143.45 | 7 | 12.2 | E2 | Soft | 8.2 | F3; T1 | Hard |
| SUMR25 | 2 | 169 | 5.9 | 9 | E2 | Soft | 6.33 | F3; T1 | Hard |
| SUMR50 | 2 | 250 | 3.95 | 4.08 | E1 | Soft | 4.08 | F3; T1 | Soft |

The SUMR-D blade being the shortest of the two-bladed designs studied exhibits a classical flutter instability at 53.98 rpm. Next, the SUMR13 series of blade designs are progressively longer and exhibit a decreasing trend in the flutter speeds. A lower torsional stiffness is generally observed in the SUMR13B blade and can be related to the lower rpm classical flutter margin observed. This shows the significance of the flapwise and torsional stiffness on the flutter margin of the blades. Further, we see much lower rpm for the edgewise instability in SUMR13B versus SUMR13A and SUMR13C, which is due to the much lower edgewise stiffness in the most slender chord SUMR13B design. The edgewise stiffness of the SUMR13C blade is significantly higher than that of the other blades because the SUMR13C blade design is driven by edgewise loads (Yao et al., 2021a) and the design was approached with both larger chord and more usage of TE reinforcement, which also helps to increase the torsional stiffness of the blade (Figure 7). The SUMR25 blades also follow similar trends to the 13MW designs where the flutter instabilities occur earlier and have a flapwise and torsional contribution. As for the SUMR50 blade design, the edgewise instability and the first classical flutter mode occur at the same speed of 4.08rpm. The lower stiffness of the blade in relation to its size contributes to this behavior of the two instability types coinciding. Looking across most of the blades, the classical flutter modes for the two-bladed designs are primarily flapwise and torsion modes as shown in Table 5. There exists some degree of edgewise contribution to the flutter mode shapes but are comparatively lower than that of the three-bladed blades due to the higher edgewise stiffness observed in the two-bladed designs. Looking at the chordwise centers for these turbines (shown in the Appendix in Figure A9-A14), it can be observed that two-bladed turbines follow a similar trend to the three-bladed designs in that the higher margins between EA and CG tend to decrease the flutter speeds.

### 4.3 Comparison of Flutter Behavior for Two- and Three-bladed Rotors

Based on the results of the prior sections (4.1 and 4.2), we can draw some conclusions from the studies of flutter behavior in both three-bladed and two-bladed rotors. In summary, we note the following: (1) flutter margins of wind turbine blades tend to decrease with the increase in blade length, (2) more innovative, optimized blades like the SNL100-03, IEA 3.4MW, SUMR13A tend to have a lower flutter margin, (3) three-bladed designs for large wind turbine blades tend to have an unstable edgewise




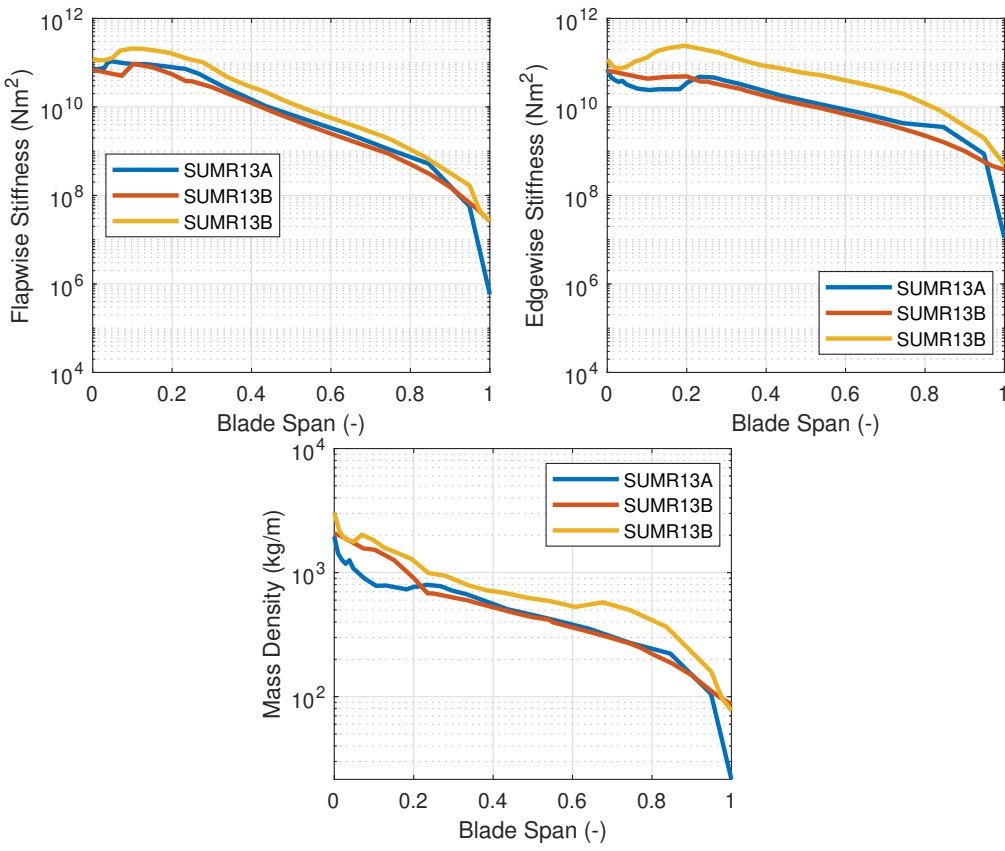

**Figure 7.** Flapwise stiffness, edgewise stiffness, and mass properties for the SUMR13 series of blades

mode that occurs before the onset of classical flap-torsion coupled-mode flutter, (4) two-bladed designs show the opposite trend and tend to have an unstable classical flap-torsion coupled flutter mode at a lower rpm with an edgewise instability occurring at higher rpm, and (5) the margin between the chordwise CG and Elastic Axis is an important design factor for the flutter speed for the turbine for both two- and three-bladed rotors.

    Figure 8 summarizes the various blade designs analyzed in this study including both two- and three-bladed designs. By
observing the classical flutter and edgewise instability margins as a function of the blade length, we see the asymptotic nature of the margins that decrease with the increase in blade length but avoid reaching the value of 1. Two- and three-bladed designs exhibit this asymptotic nature of margins as seen in Figure 8. This asymptotic nature of flutter margins is primarily due to the decrease in rated operating speeds of the turbine with the increase in blade length, and the inherent increase in structural stiffness needed to withstand the loads as the rotors get larger. Similar results were also observed by Kelly et al. (Kelley and
Paquette, 2020) where a scaling study for large blade designs was conducted and it showed that the flutter speeds are asymptotic with blade length. But, this is the first study that observes the asymptotic nature of these two- and three-bladed rotors using a large set of real designs based on detailed structural designs that meet requirements of international design standards for





strength, deflection, fatigue, buckling, and dynamic stability. Next, looking at the unstable edgewise modes, we can observe a correlation with the edgewise stiffness of the blades. For most of the three-bladed designs, the edgewise instabilities are below
the classical flutter modes and for the two-bladed designs, they are higher than the classical flutter modes. We can also observe that they follow a similar asymptotic trend to the classical flutter modes. It is important to note that the observed classical flutter modes are "hard" flutter modes, where the change in the damping ratio into the unstable region is sharp or sudden whereas the edgewise instabilities tend to have a "soft" or shallow transition into the unstable region.

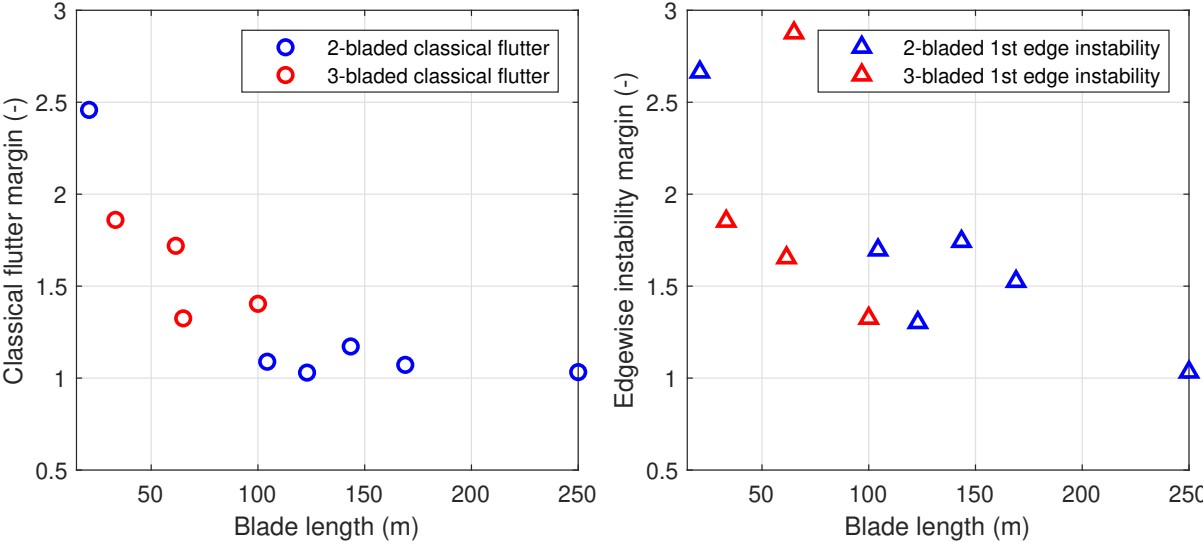

**Figure 8.** Classical flutter and edgewise instability margins for various wind turbine blade series were analyzed. The blue markers are three-bladed turbines, the red markers show the two-bladed downwind SUMR series of blades.

## 5   Blade Redesign to Improve Flutter Margin

As shown in the prior section, flutter instability of both types (edgewise vibration and classical coupled-mode flutter) are a concern for highly flexible, large-scale rotors, especially beyond 100m in length. A key question we examine now is how to address or mitigate flutter in the wind turbine design process. The fundamental list of options or approaches for flutter mitigation is three-fold: (1) pursue aerodynamic redesign solutions by changing; for example, the airfoils or chord design, (2) pursue a new control strategy such a flutter detection and flutter suppression, or (3) pursue a structural redesign solution to
adjust blade mass, stiffness, and chordwise CG properties. Of course, the most desirable design solution is to have a wind turbine blade design that has a higher, improved flutter margin while constraining blade mass and reducing system cost. In this section, we investigate the third option, which is a passive design solution to mitigate flutter in the structural design process. But, what is the best approach to improve flutter speed through a blade structural redesign? To answer this question, a design study is carried out to investigate the impact of material placement and selection on the flutter margin using a tool called



AutoNuMAD. The effect of the material placements on the flutter speed is analyzed to better understand how blades can be developed to have higher flutter margins and also give a precursor to implementing structural optimization.

## 5.1 AutoNuMAD and Design Space Exploration

AutoNuMAD (Chetan et al., 2019a) is a wind turbine blade design tool developed at the University of Texas at Dallas and is based on the NuMAD framework created by Sandia National Labs (Berg and Resor, 2012). This tool simplifies the process
of wind turbine blade design by allowing the user to define design variables and to manage all the detailed and complicated blade design information including airfoil geometry, the variation of chord and twist, and detailed composite material layups. AutoNuMAD offers built-in features to perform additional analysis, including a bill of materials, manufacturing cost analysis, modal analysis, Campbell diagrams, and flutter prediction. Additionally, the tool allows the user to run analysis codes like OpenFAST (Jonkman, 2020), MLife (Hayman and Buhl Jr, 2012), and MExtremes (Hayman, 2015) within the framework to
be able to use the desired outputs in an optimization loop. Figure 9 covers all the aspects of blade design and optimization under a unified framework – all features that are essential for the design of blades. AutoNuMAD utilizes well-established optimizers from the MATLAB Optimization and Global Optimization toolbox. The flexibility to vary the blade structural parameters allows the user-defined Monte Carlo sweeps to explore a design space or find a stable starting point for further optimization.

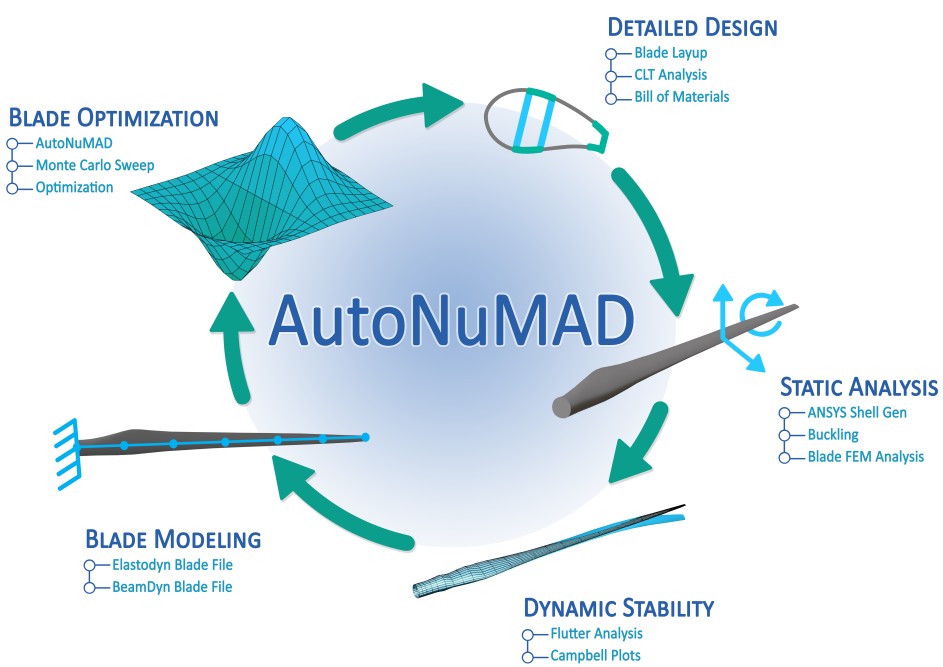

**Figure 9.** AutoNuMAD Framework





### 5.2 Structural Design Mitigation: Trends for Improving Flutter Speeds of Wind Turbine Blades

In this section, a trends study is carried out on the three-bladed SNL100-03 (Griffith and Richards, 2014) design and the two-bladed SUMR13C (Zalkind et al., 2019; Yao et al., 2021a) blade design. The flutter mitigation solution via structural design is approached by varying the leading edge (LE) and trailing edge (TE) reinforcement in the blade structural design (as illustrated in the cross-section of Figure 10). The choice of LE and TE reinforcement to address the flutter speed is a judicious choice as the placement of the LE and TE in the cross-sectional geometry contributes primarily to the edgewise stiffness, torsional

stiffness, chord-wise center of mass, and chord-wise elastic center. As observed in previous sections all these properties have a strong contribution to the flutter margin of the blade and hence the LE and TE reinforcement plays role in determining the flutter margin of the blade.

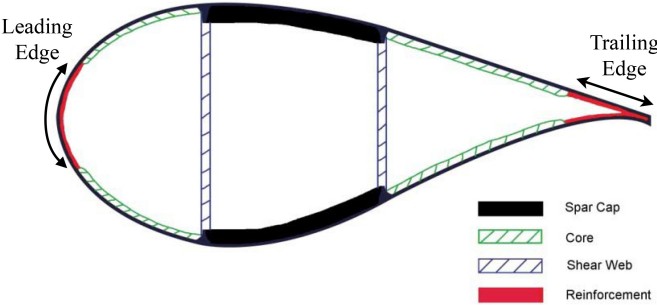

**Figure 10.** Cross-section of a wind turbine blade with two shear webs (Griffith and Ashwill, 2011)

We begin by examining the SNL100-03 blade for a three-bladed rotor. In this study, we vary the LE and TE reinforcements to study the ideal usage of these layers to influence not only the flutter speed, but also the overall blade mass, blade cost, and

blade stiffness properties. Thus, we vary the LE and TE reinforcement by scaling the existing number of plies by a scaling factor. For the LE reinforcement, because the SNL100-03 blade does not have LE reinforcement, we define 12 layers of LE reinforcement as the maximum of the range to be consistent with the range of the number of layers of TE reinforcement for this study. Figure 11 shows us contour plots of the various resulting flutter properties for the SNL100-03 blade. Note the red star which indicates the SNL100-03 blade, which is the baseline for this LE and TE reinforcement sweep study. From these plots

we can observe that (1) flapwise stiffness is more sensitive to variations in LE than TE reinforcement, (2) the torsional stiffness is also more sensitive to variation in LE at higher TE layers; and it is more sensitive to variation in TE at higher values of LE, (3) flapwise frequencies are mostly sensitive to the variation of LE layers than TE, (4) in terms of classical flutter speed, it is most sensitive to variations in LE than TE reinforcement. The sensitivities equalize at higher values of TE reinforcement, (5) the sensitivity of flutter speed to TE at higher TE values is primarily driven by stiffness increases, (6) the increase in sensitivity

of flutter speed to LE at lower TE can be explained as driven by the CG location in addition to the stiffness effect, and (7) the maxima for the flapwise and torsional stiffness occur at higher LE and TE, but for the flutter speed, this occurs at lower TE and





higher LE reinforcement. From these observations of the trends study we note the benefits of additional LE layers to increase the flutter speed versus the TE layers. Similarly, we can have blade designs that are lighter than the baseline structure but have a higher flutter speed.

**Figure 11.** LE and TE sweep for the three-bladed SNL100-03 blade. The red star in the plots indicates the baseline blade design (SNL100-03).

Next, we examine a structural redesign for a two-bladed rotor for the SUMR13C blade. Here, the number of layers for both TE and LE is scaled by a factor from 0 to 2 from the baseline design having scale factors of 1. The resulting contours are shown



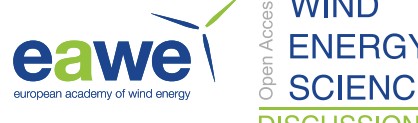

**Figure 12.** LE and TE sweep for the two-bladed SUMR13 blade. The red star in the plots indicates the baseline blade design (SUMR13C).

in Figure 12. A few observations are noted from these results as follows: (1) flapwise and torsional stiffness are more sensitive to variations in TE than LE, (2) for a lower number of TE layers and higher LE layers, the classical flutter speed is sensitive to variations in TE, (3) in general, the flutter speed is more sensitive to the variation of TE. This is mostly driven by the change in the CG than the increase in stiffness as evident by the Figure 12, (4) the first torsional frequency of the blade remains relatively the same across the variation of TE and LE, and (5) from understanding the design space, we can increase flutter speed for the




baseline blade by increasing the LE reinforcement and reducing or not modifying the TE layers. In contrast to the trends for the SNL100-03 blade (Figure 11), the SUMR13C is more sensitive to changes in TE. This is primarily due to the use of carbon in the SUMR13C, whereas glass fiber is used in the SNL100-03 design.

From analyzing the design space for both the two- and three-bladed designs we understand that the design space for each blade is important as the sensitivities tend to be unique. The design space provides a design direction that could lead to an optimal blade design which has higher flutter speeds. It is important to identify the flutter instability within the design or optimization framework given the prominence of flutter in the design of highly flexible, large-scale rotors. This is especially true when edgewise instabilities as seen in the three-bladed designs are present. Additionally, if the modes are not correctly

identified, the jumps in the different unstable modes could lead to discontinuities for gradient-based optimization systems. To overcome this, techniques like MAC and other reliable mode tracking methods must be implemented (McDonnell and Ning, 2021; Chen and Griffith, 2021).

## 6    Concluding Remarks and Future Work

From the study of flutter in wind turbine blades, a pattern emerges of larger, highly flexible blades having lower per-rev flutter

speeds. There is a need to better understand the flutter behavior of large wind turbines, including new configurations such as two-bladed rotors; and there is a need to optimize rotor designs to avoid flutter while still meeting design performance and cost constraints. This is the focus of this work. Toward these goals, the main contributions of this paper are (1) a comprehensive study of the flutter behavior (e.g.; flutter speeds, flutter mode shapes) for a series of two-bladed wind turbine rotors, (2) a comparison of flutter behavior for both two- and three-bladed rotors including an examination of classical coupled-mode

flutter and edgewise vibration, (3) the observation of asymptotic nature of wind turbine flutter speeds with respect to blade length, and (4) a trend study of LE and TE reinforcement to mitigate flutter through structural design of material choice and material placement.

Firstly, we performed a comprehensive study of the trends in flutter characteristics as they vary with structural design choices (materials, geometry) for a series of wind turbine blades. The first observation is that flutter speeds are reduced as blade length

increases. Next, the lowest RPM unstable modes in three-bladed turbine blades tend to be edgewise in nature, which is then followed by the classical flutter modes which are flap-torsion driven. These are primarily observed in three-bladed designs due to their lower edgewise stiffness resulting from their slender nature. For two-bladed designs, the first unstable mode contributions were primarily flap-torsion coupled mode distinctive of classical flutter. An interesting finding is that edgewise flutter is effectively eliminated with two-bladed rotors as these designs inherently have a larger chord, and as a result have

higher relative edgewise bending stiffness. Additionally, in examining a large number of wind turbine blades we observe an asymptotic nature of the flutter margins with respect to the size of the wind turbine blades. This points to requiring flutter to be actively considered as a constraint in the design of wind turbines and wind turbine blades.

Finally, the trends showed the need to increase flutter margins in large blade design. Thus, a detailed trends study was performed on a three-bladed SNL100-03 blade and a two-bladed SUMR13C blade to understand the effects of placing LE



and TE reinforcement along the blade. The results show how a careful combination of LE and TE reinforcement can result in significant increases in flutter margins compared to the baseline blade while maintaining/reducing the blade mass. The increases in flutter margins are primarily observed when the flutter modes that interact are of higher frequency. For the SNL100-03 blade we observe that the flutter speeds are more sensitive to the leading edge layers whereas for the SUMR13C the flutter speeds are more sensitive to the trailing edge reinforcements. The difference between the two trends is driven by the SNL100-03

having LE reinforcements made of glass-fiber; but the TE reinforcements for the SUMR13C are made of carbon fiber which is significantly stiffer. In both cases, we observe that it is possible to increase the flutter margins while maintaining or reducing the mass of the blade.

A future extension of this work should include a comparison of these flutter predictions with time-domain aero-elastic simulation codes capable of modeling key blade dynamics. Regarding the flutter prediction tool, we found that our tool is

in very good agreement with other works, which we determined from a benchmarking study. We investigated the still air assumption of the flutter tool and, based on other works and our own studies, it is found that the still air assumption is a reasonable assumption having a small yet conservative effect on flutter speed prediction. Further, we envision the design tends study to be used as a basis to optimize the turbine designs. This can be in the form of a constraint while optimizing the turbine cost or to optimize the blade for flutter instability as a part of a co-design process. It is to be noted that for gradient-based

optimization methods the modes have to be tracked carefully due to the nature of how flutter modes switch with the changes in the blade design.

*Author contributions.* MC contributed with Conceptualization, Investigation, Methodology, Software, Visualization, and Writing – original draft preparation. SY contributed with Conceptualization, Software, and Writing – review & editing. DTG contributed with Conceptualization, Funding acquisition, Supervision, Writing – review & editing.

*Competing interests.* The authors declare no conflict of interest.

*Acknowledgements.* The research presented herein was funded by the US Department of Energy Advanced Research Projects Agency – Energy under the Segmented Ultralight Morphing Rotor project (award DE-AR0000667). Any opinions, findings, and conclusions or recommendations expressed in this material are those of the authors and do not necessarily reflect the views of ARPA-E. The authors are grateful for the support of the ARPA-E program and staff.





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

## Appendix A: Two- and three-bladed turbine blade planforms



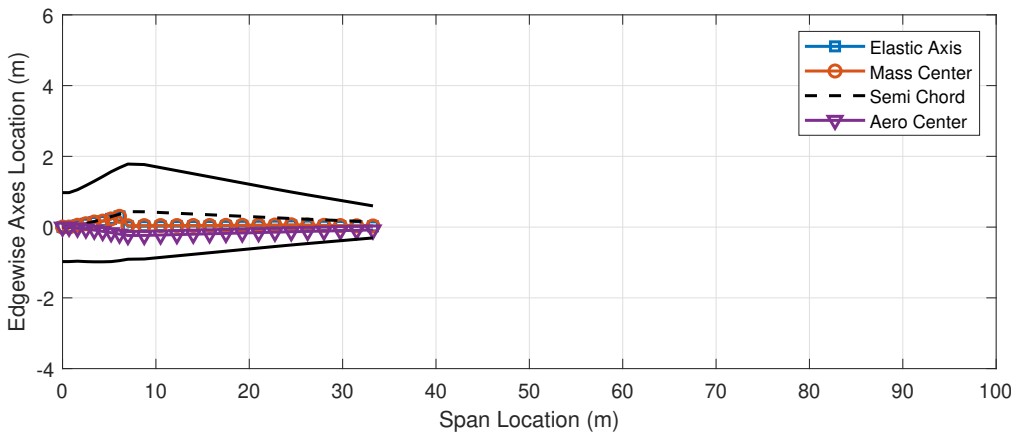

**Figure A1.** Planform for the WindPACT 1.5MW turbine blade

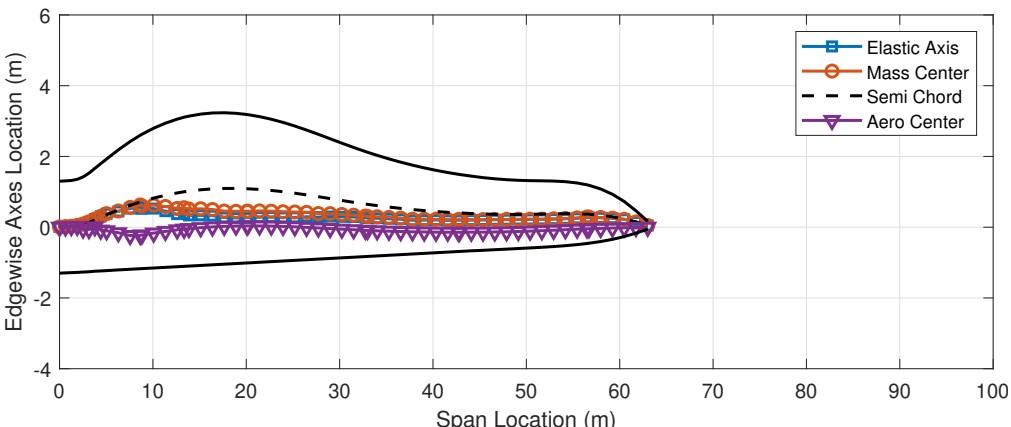

**Figure A2.** Planform for the IEA 3.4MW turbine blade

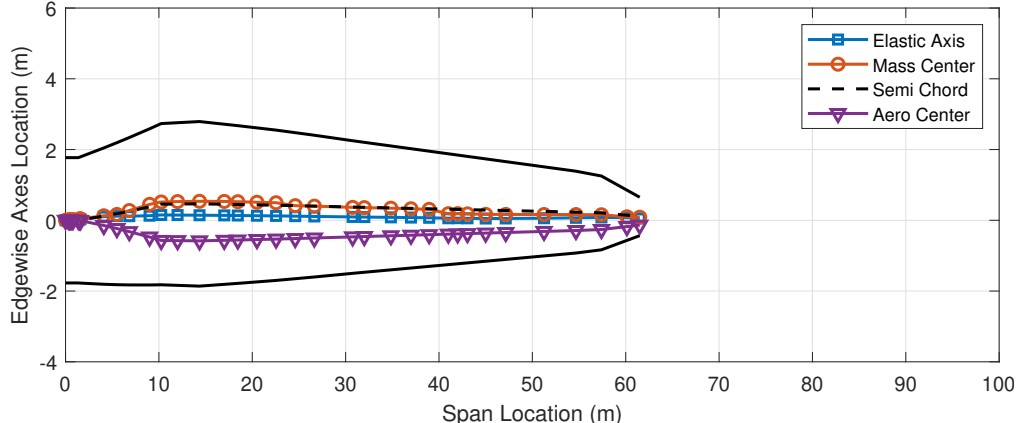

**Figure A3.** Planform for the NREL 5MW turbine blade



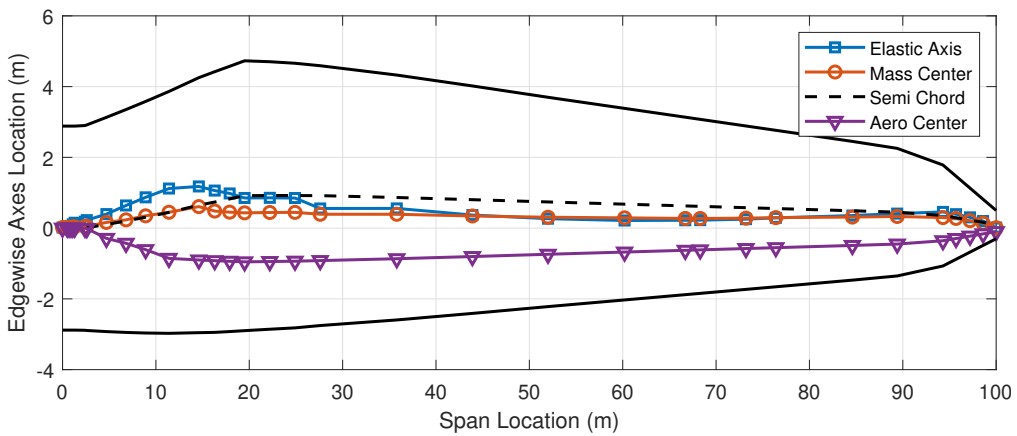

**Figure A4.** Planform for the SNL100-00 turbine blade

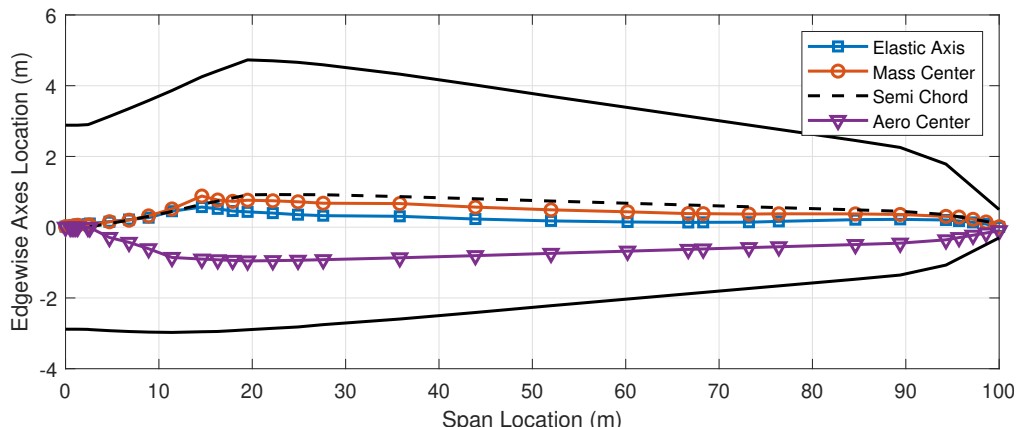

**Figure A5.** Planform for the SNL100-01 turbine blade

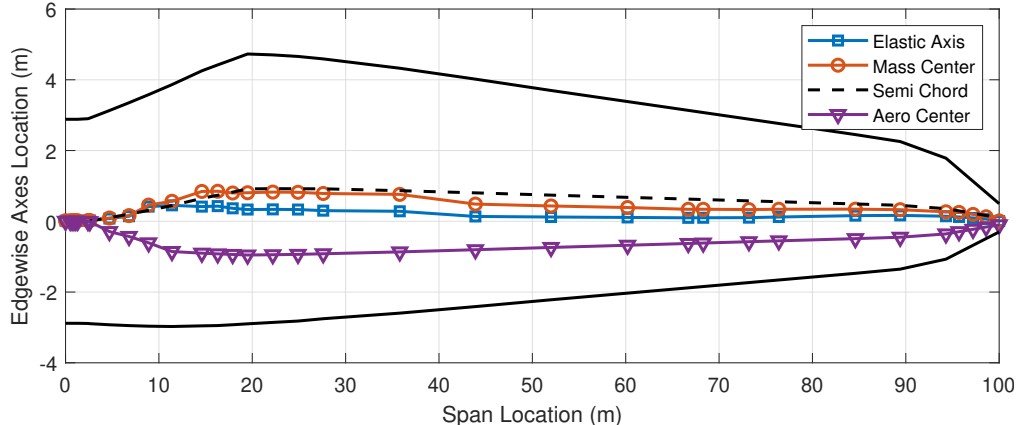

**Figure A6.** Planform for the SNL100-02 turbine blade

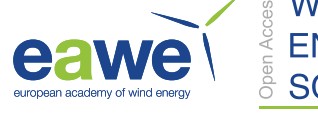

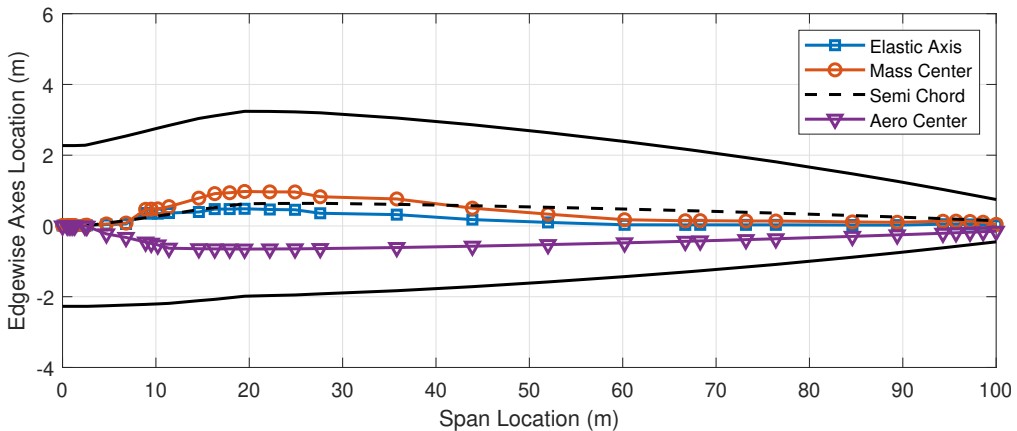

**Figure A7.** Planform for the SNL100-03 turbine blade

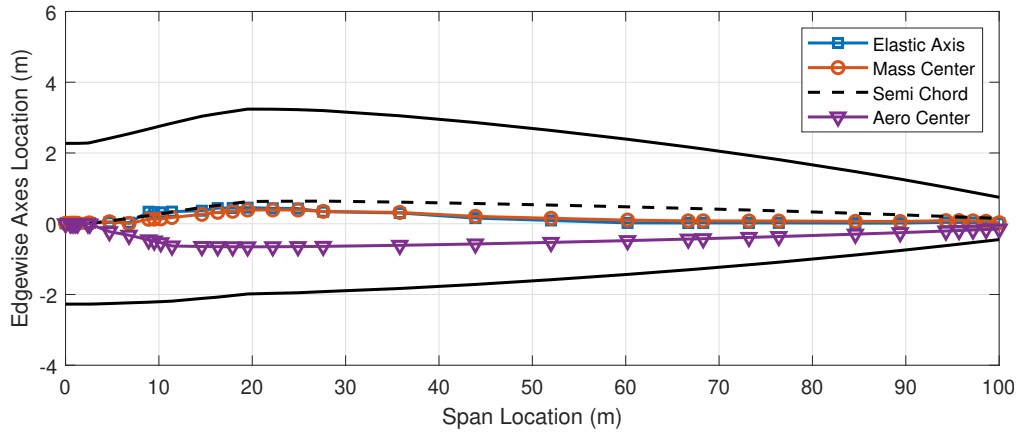

**Figure A8.** Planform for the UTD100-04 turbine blade

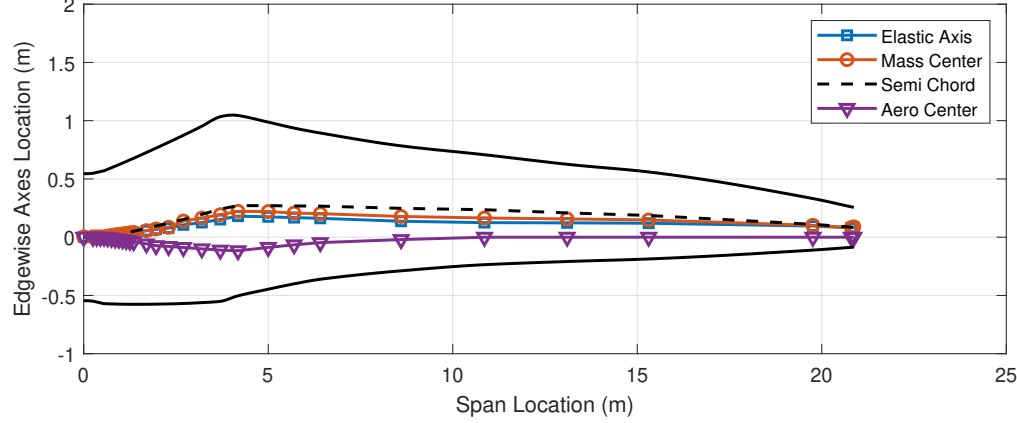

**Figure A9.** Planform for the SUMR-D turbine blade



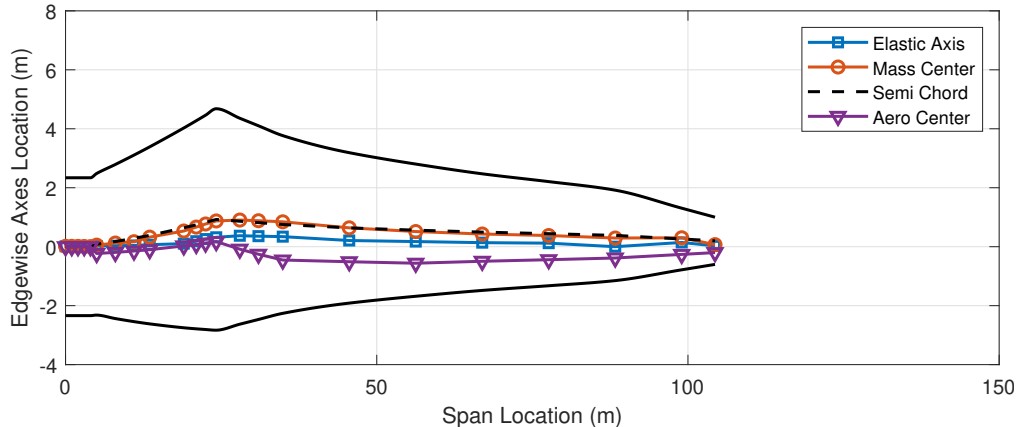

**Figure A10.** Planform for the SUMR13A turbine blade

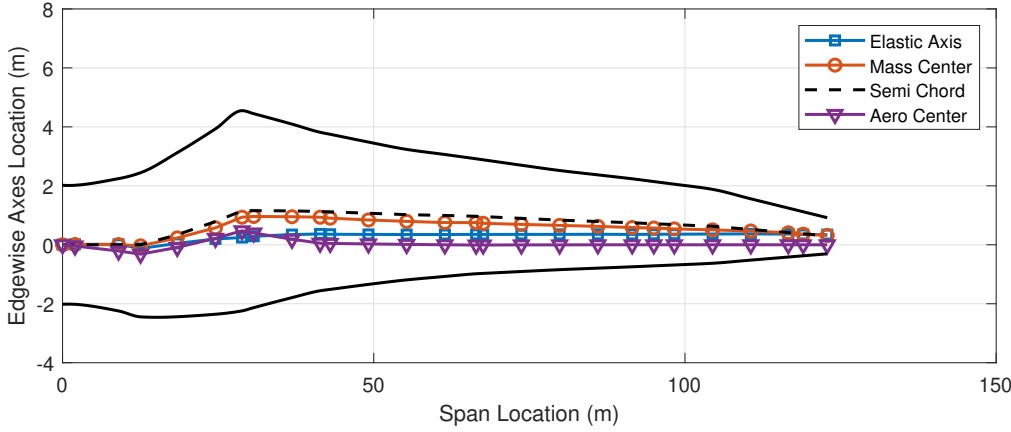

**Figure A11.** Planform for the SUMR13B turbine blade

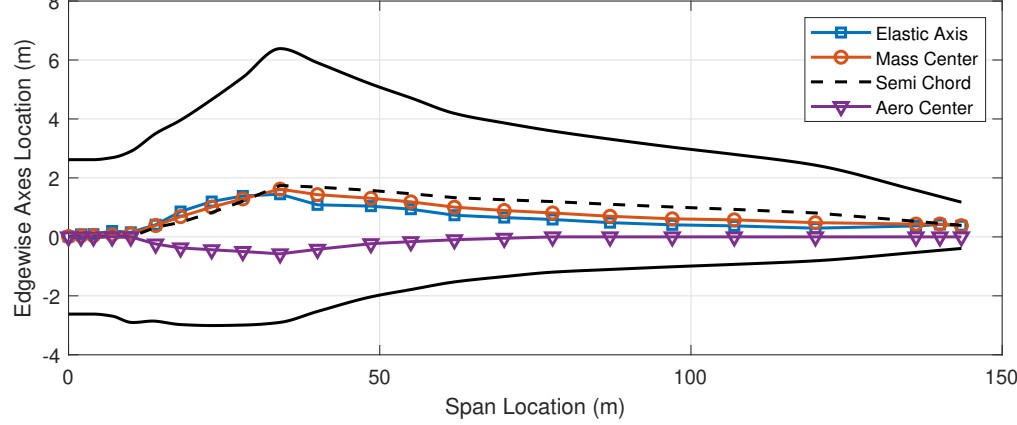

**Figure A12.** Planform for the SUMR13C turbine blade



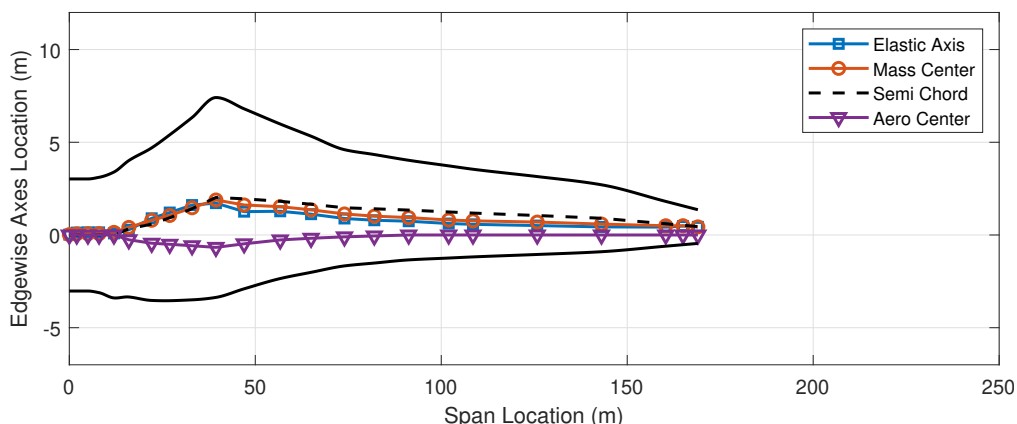

**Figure A13.** Planform for the SUMR25 turbine blade

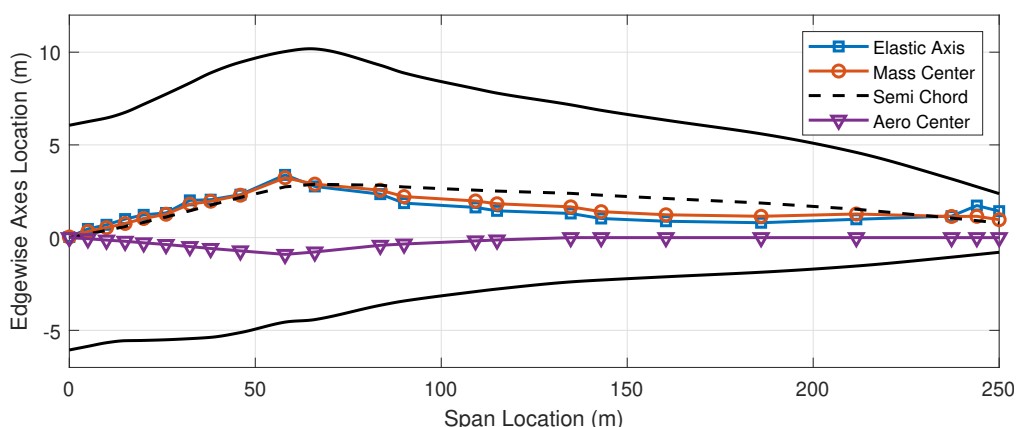

**Figure A14.** Planform for the SUMR50 turbine blade