# Peer review of "Flutter Behavior of Highly Flexible Blades for Two- and Three-bladed Wind Turbines"

_Wind Energy Science, 2021_

## Referee Comment (RC2)

Review of the paper entitled: "Flutter Behavior of Highly Flexible Two- and Three-bladed Wind Turbine Rotors" by M Chetan et al.

General comment

The paper deals with a remarkably interesting analysis of the flutter margins for wind turbine blades of different lengths. After analyzing the flutter margins of existing blade designs, the Authors proposed a re-design methodology aimed at improving the flutter margins themselves.

The tools used for estimating the flutter onset (Theodorsen unsteady aerodynamic theory and p-k method for the computation of blade eigenvalues) are suitable for the scope of the work.

The paper is well-written and, mostly, enjoyable to read. It contributes to state of the art in this field and has good citing potential.

This said, I have a list of issues that should be addressed before that final acceptance of the paper (see "important comments"). Finally, there are also some minor comments, that the Authors should also consider in the amended version of the manuscript.

My recommendation is "minor revisions."

Important comments

1. The title of the paper creates an extremely high expectation. In fact, reading it, one may think that the work deals with a flutter analysis of the entire rotor also including hub and shaft (and in general an even simple flexible supporting structure). In the end, the analysis considers only the isolated blade. This means that only the mutual interaction of the blade modes is analyzed in the work. Other possible flutter sources, e.g., those coming from interactions blade, tower, and whirling modes, are not considered, since whirling modes exist only if the supporting structure of the rotor is assumed. Moreover, the words "three-" and "two-bladed" can lead readers to expect an analysis which considers different hub-typology for the two-bladed configuration (for example with a teeter hinge). All these issues should be fixed and, accordingly, I request the Authors to slightly modify the title.

2. Directly connected to the previous point: it could be important to declare the typology of the hub (hinged or fixed teeter) of the analyzed blades. This could lead to a deeper interpretation of the results. Given the status of the paper, the two-bladed configuration differs from the three-bladed one only for the longer length of the blade. From this point of view, Figure 8 shows a clear trend in the flutter margins that depends on the sole length of the blade, whereas a different behavior between the two rotor configurations is not visible.

3. In the redesign section, it should be fair to say that the blades are not redesign with the same constraints of the nominal one. Hence, all design constraints (maximum tip deflection or fatigue) should be verified after the redesign.

Minor comments:

1. Line 5/Abstract: "two-bladed rotors" and "downwind configurations" are both listed as "new rotor concepts". They are not new, but rather "unusual". Please, modify the sentence.

2. Line 111: I think that the name of the flutter computation process is "p-k method" and not "p-f". I could be wrong but, in any case, it is important to indicate the chapter and the section where the method is illustrated in Wright's and Cooper's book.

3. Line 327: "From these observations of the trends study we note the benefits of additional LE layers to increase the flutter speed versus the TE layers". This is something that could be also foreseen, as a forward motion of the sectional center of gravity is typically beneficial in terms of flap-torsion flutter speed. But my question is: Do Authors think that these results depend on the forward motion of the sectional gravity center or on other structural reasons?

4. Line 328: "Similarly, we can have blade designs that are lighter than the baseline structure but have a higher flutter speed". This sentence may be misleading. In fact, the re-design consists in a re-arrangement of the internal blade layout, while other design constraints are not verified (e.g., maximum tip deflection, fatigue etc.…). Hence, it is hard to say that a lighter design is obtain because the process is performed using the very same constraints of the nominal blade. Please, clarify or remove the sentence.

5. Line 377: "it is possible to increase the flutter margins while maintaining or reducing the mass of the blade". Here again, the sentence is clear but prone to misinterpretation as written in the previous comment.

6. Conclusions: if relevant, the study of the flutter of the entire turbine could be mentioned within the possible extensions of the present work.

---

## Author Comment (AC1)

Author Response: Reviewer #1

**Reviewer Comments in black text and author response in red text.**

Dear authors, thank you for a nicely written article on a very relevant topic. The paper reads well and I recommend publishing it, pending some key improvements.

The authors would like to thank the reviewer for taking the time in reviewing the article. Flutter and aero-elastic instability in general are a critical failure criterion for large wind turbine blades, and we believe our work supports this.

The biggest feedback is that the article falls short of what I was hoping to find in it. First, the paper is a little blurry on what is novel in terms of theory and models. I am confused whether the models presented in section 2 are any different than the models available in literature and correctly cited in the bibliography. This should be made clearer.

The theory implemented is similar to the ones available in literature. The theory is presented to present the approach in brief form and to make clear the implementation of the theory compared to that available literature thus we feel that more details aren't warranted given this summary and citation of relevant other works.

In addition, the exploration of the design space is very limited. Why only LE and TE reinforcement thickness? What about outer shell skin thickness, or spar cap placement, or relative placement of LE and TE? Overall the reader is left with many unanswered questions, and this is a pity, since you have all the tools to answer these questions.

This is an interesting observation; however, the primary focus of this paper is on (1) an introduction to flutter analysis of two-bladed rotors, (2) a critical comparison of two- and three-bladed flutter behaviors, their instability trends, and asymptotic nature of flutter, (3) investigation of the leading-edge and trailing-edge re-design motivated by the spanwise properties in #2, and (4) recommendations for a more comprehensive approach to wind turbine blade solutions to mitigate flutter. The design space exploration was limited to the TE and LE thicknesses due to their strong influence on the flutter margins and blade mass. Yes, the tools available to us can be used to explore the effect on flutter due to other structural components of the blade. This was outside our scope given for this work, and we note that future work should examine other structural design approaches.

Finally, I would like to see what it takes to get the designs up to an acceptable flutter margin. It would be good to define this acceptable limit rather than increasing the margin to an arbitrary amount. Then it would be good to see the feasibility of the blade from a design/manufacturing/cost perspective.

We do set a flutter margin of 1.2 as a limit (Zalkind, WES 2019), we do agree showing this on the contour plots (contour lines have been updated to show the flutter margin of 1.2) and providing potential designs will improve the understanding of the design space exploration. The detailed feasibility study of the blade models is outside the scope of this LE/TE parameter study which has an objective to provide a trends study for blades with higher flutter margin with minimal mass increase. We note potential future work extensions in the conclusions section.

Below you find a longer list of comments pointing to specific parts of the text that have margin for improvements:

**Title**: the title does not seem to be the most accurate: none of the 3-bladed rotors is highly flexible. On the contrary, the 3-bladed designs are actually fairly stiff

We have modified the title to be more specific and address concerns of the reviewers with new title of "Flutter Behavior of Highly Flexible Blades for Two- and Three-bladed Wind Turbines". The study looks at blades from 21-meters to 250-meters in length, with the title we aim to note that the paper mainly focuses on examination of larger blades that are highly flexible in nature.

**Figure 2**: This is a nice image, but it is slightly confusing. It reports the WindPACT rotor, which is not listed in 3.1. The IEA10 is instead listed in 3.1, but is not included in Figure 2 nor in Table 1

Thank you for the comment. We have corrected Section 3.1 to reflect the correct turbines used in this study by adding the WindPACT rotor to the list. Figure 2 has also been updated to show the turbine blade lengths.

**Table 1**: columns on the right miss units

Thank you for the comment; we have corrected Table 1.

**Table 2**: wouldn't it be more appropriate to use the digital twin described in https://doi.org/10.1002/we.2636?

To be consistent with all the other turbine models used we determined that the "final design" of the SUMR-D (pre-fabrication model) would be the best version of the model to use; thus the choice in this case.

**Page 8**

line 168: please quantify "slightly higher"

Section 3.3 has been updated with specific numerical values.

Line 170: I think it's a little hard to make this claim without evidence, especially since the flap and torsion are coupling in this instability.

Agree. The sentence has been removed from the manuscript.

Line 172: don't all these references use the same exact formulation? I find this misleading throughout the paper, see points above

Owens, Griffith, and the current study use similar formulations, but Hansen and Pourazarm are different. This was carried out to verify the implementation of the flutter prediction tool.

**Table 3**: It might be nice to add the rated rpm for each design

Table 3 has been updated to include rated rpm.

**Figure 6 and 7**: please add torsional stiffness

Figures 6 & 7 have been updated to include torsional stiffness.

**Page 12**

Line 227: solidity of the rotor, not of the turbine

Thank you, it has been updated.

Line 228: "larger margins between the EA and CG locations in the chordwise direction." This is not necessarily true, rather an artifact of some simple scaling steps

The blade layup designs for the SNL100 series and the SUMR13 series are carried out at the composite layup level to meet design requirements like max strain and deflection. The location of the EA and CG are a result of this detailed composite layup design process and not due to simple scaling.

**Table 5**: 13A and 13B have a super low margin. Perhaps these (and the SUMR50) cases would be most suitable for additional analysis in the next section since they have the lowest margin. Also, the flutter margin of SUMR50 is much too low to be a reasonable design. The authors should make some note of this.

The SUMR13A & SUMR13B designs are intermediate designs. In the SUMR13 series the SUMR13C design is considered as the final version hence it was selected to represent the two-bladed designs in Section 5. Yes, the SUMR50 wind turbine blade design has a significantly low flutter RPM, and more work could be done to improve the flutter margin in future work. Due to the maturity of the SNL100-03 and the SUMR13C designs they are used in the design space study carried out in Section 5. A future extension of the work could include optimizing the intermediate and poorly performing designs such as the SUMR50 and the earlier SUMR13 models. Section 4.2 has been updated to reflect this.

**Page 14**

line 262: what about the aerodynamic center as an important design driver?

Yes, we agree that AC is an important factor it was included in Section 4.3, paragraph 1, bullet point #5.

Line 272: "real designs", This is confusing. These aren't real designs except for the SUMR-D blade, which is a blade designed with scaling laws, so not at all close to industrial products.

By "real designs" we intended to express that the blades analyzed in this study were not purely scaled designs but rather were detail designed to meet a comprehensive set of standards-based design requirements. The manuscript has been updated to explain this better, and the term "real designs" has been dropped.

**Page 15**

Line 273: I question the ability of the SUMR-50 blade to meet these requirements given the low flutter margin.

A note has been added to Section 4.2 to reflect the above.

**Page 16**

Section 5.1 would benefit a revision. This study doesn't use any of the (very nice) features listed in the text, while it would be more useful to know how the blade structures and aerodynamics are modeled

Section 5.1 re-introduces the AutoNuMAD tool. We have updated the section to provide a better context of what specific AutoNuMAD features are used in this current paper.

Line 298: I wonder if these analyses consider the blades to be rotating

Yes, modal analysis, Campbell diagram and flutter prediction tools all include rotational effects.

Figure 9: please focus on the aspects related to this study. For example, how have you been computing blade stiffnesses? With PreComp? Or BPE? Or maybe Becas?

Section 5.1 re-introduces the AutoNuMAD tool. We have updated the section to provide a better context of what specific methods are used in this current paper for computing blade stiffnesses.

**Page 17**

Line 308: I was hoping to understand much more about the solution space. For example, what about the effect of chordwise placement of spar caps and shear webs. And what about chordwise placement of the blade axis? This is only one of many design choices that blade designers need to take. The feeling is that you've not moved the needle on this design exploration to a sufficient amount. Much can and should be done.

The current paper focuses on (1) a comprehensive study of flutter behavior of two- and three-bladed wind turbine rotors, (2) comparison of classical coupled-mode flutter and edgewise vibration, and asymptotic nature of these observations with blade length. The scope of the design space exploration was limited to the LE and TE due to flutter margin's sensitivity the LE and TE. We agree that a future continuation of this work should include larger number of blade components (e.g.; spars, skins) to more fully explore the design space to mitigate flutter and edgewise vibration, but this is outside the scope of this work.

Line 309: Related to the point above, outer and inner skins are much more important for the torsional behavior than LE and TE reinforcements.

Yes, we agree that the skins have a significant effect on the torsional stiffness of the blade cross section. The LE and TE were chosen since they have a more significant impact on the flapwise stiffness, edgewise stiffness, EA and CG in addition to torsion.

Line 317: One tricky aspect here is that you're likely using shell elements in modeling of the blade structure. Shell elements accept any thickness, although fin-shape trailing edges cannot physically fit many composite layers

Yes, the PreComp cross sectional analysis tool used in this study does consider the laminates to be in some cases moderately within the outer mold line of the cross-section. This can be avoided by using a finite element based cross sectional analysis tool like BECAS or ANBA4.

Line 320: point #1 is kind of obvious... because of the airfoil shape, moving away from the LE moves you from the neutral axis faster than at the TE. LE is therefore better suited to increase flap stiffness, although this is probably not very relevant. You don't add material at the LE to increase flap stiffness

Yes, we agree that adding material to the LE is the first choice to increase the flapwise stiffness of the blade. But of the two, LE is the better option.

Line 322: flapwise frequencies are more (not mostly) sensitive

Thank you, we have corrected this in the manuscript.

Line 323: point #4 seems just a natural consequence of points #1 and #3

Yes, that is correct.

Line 326: please explain the sentence "for the flutter speed, this occurs at lower TE and higher LE reinforcement"

Here we point out that the maximum flutter margin in our study occurs when the number of LE layers are higher, and the number of TE layers are lower. The sentence has been re-written to make it clear for the reader.

**Figure 11** in the top left corner I'd recommend plotting flutter margin. In the bottom right, it's unclear why the space is so nonlinear

Figures 11 & 12 have been updated to show the flutter margins instead of flutter speed.

**Page 20**

Line 339: I have the feeling that what's truly happening here behind the hood is a change in the positions of EA and COG. I'd find that analysis much more informative than a parametric study on LE and TE thicknesses. It's hard to draw any general conclusion out of this parametric study

Yes, we do agree that the change of EA and CG which is most sensitive to variation in TE and LE is causing the flutter margins to change. A detailed study of the EA and CG would definitely be valuable to understand the designs space but is out of the scope of this current study.

Line 346: please spell out acronym MAC

This has been updated in the in Section 5, last paragraph of the manuscript.

Line 349: The analysis seemed to show some correlation for "longer" blades, should use that instead of "larger"

This has been updated in Section 6 of the manuscript.

Line 350: rotor instead of turbine

This has been updated in Section 6 of the manuscript.

Line 352: The difference between points 1 and 2 is subtle and having a single point would improve readability

This has been updated in Section 6 of the manuscript.

Line 359: flutter margins are reduced as well, which is likely more important than flutter speeds

'Flutter speed' has been replaced by 'flutter margin' in the manuscript.

Line 366: This was already said in line 355. The whole paragraph seems convoluted and could be shortened focusing on the key conclusions, which I understand as:

1) Flutter margins decrease with blade length

2) 3-bladed rotors show edge instabilities at lower rpm than flap-torsion flutter

3) The opposite holds true for 2-bladed thanks to larger chords generating higher edge stiffness

Line 355 enumerates the main contributions of the paper, and the following paragraph provides a conclusion to the findings. The paragraph has been rewritten to shorten the repeated points.

**Page 21**

Line 380: All these works use the same theory. This seems to me a point of great confusion. The methods for flutter prediction used in this work seem no different than the works cited from literature. It is certainly reassuring to see that the results presented in this work match older results, but the paper should make clear that the theory is not a novel contribution, only the parametric studies and the comparison 3-vs 2-bladed rotors.

Yes, the theory used in this work is not new for flutter prediction, nor has it been portrayed as new. The theory was presented to show the different aspects that are considered in the model to explain the methods implemented in our tool. Additionally, the benchmarking was done to ensure that our flutter analysis tool was in good agreement with literature.

Line 382: I don't think your paper supports this conclusion, aside from the findings already available in literature

The details regarding the still-air assumption investigation have not been presented in this work. Therefore, the paragraph has been updated by removing this sentence regarding the still-air assumption.

---

## Author Comment (AC2)

**Author Response: Reviewer #2**

**Reviewer Comments in black text and author response in red text.**

Review of the paper entitled: "Flutter Behavior of Highly Flexible Two- and Three-bladed Wind Turbine Rotors" by M Chetan et al.

General comment
The paper deals with a remarkably interesting analysis of the flutter margins for wind turbine blades of different lengths. After analyzing the flutter margins of existing blade designs, the Authors proposed a re-design methodology aimed at improving the flutter margins themselves.
The tools used for estimating the flutter onset (Theodorsen unsteady aerodynamic theory and p-k method for the computation of blade eigenvalues) are suitable for the scope of the work.
The paper is well-written and, mostly, enjoyable to read. It contributes to state of the art in this field and has good citing potential.

This said, I have a list of issues that should be addressed before that final acceptance of the paper (see "important comments"). Finally, there are also some minor comments, that the Authors should also consider in the amended version of the manuscript.
My recommendation is "minor revisions."

The authors would like to thank the reviewer for taking the time in reviewing the article, and we appreciate the feedback provided.

Important comments
1. The title of the paper creates an extremely high expectation. In fact, reading it, one may think that the work deals with a flutter analysis of the entire rotor also including hub and shaft (and in general an even simple flexible supporting structure). In the end, the analysis considers only the isolated blade. This means that only the mutual interaction of the blade modes is analyzed in the work. Other possible flutter sources, e.g., those coming from interactions blade, tower, and whirling modes, are not considered, since whirling modes exist only if the supporting structure of the rotor is assumed. Moreover, the words "three-" and "two-bladed" can lead readers to expect an analysis which considers different hub-typology for the two-bladed configuration (for example with a teeter hinge). All these issues should be fixed and, accordingly, I request the Authors to slightly modify the title.

Thank you for the detailed feedback. Yes, the focus of this work is limited to the flutter stability of an isolated blade in still air. This assumption allows the blade designer to concentrate on the detailed design of the blade while still having a quick tool to compute the flutter margins of the blade. A full turbine flutter prediction tool introduces a large number of degrees of freedom that identification of the actual flutter modes become a more difficult task. Two- and three-bladed designs are a focus of this study to provide an understanding of the difference at a detailed blade design level. Following the suggestions by both reviewers, the title has been changed to "Flutter Behavior of Highly Flexible Blades for Two- and Three-bladed Wind Turbines".

2. Directly connected to the previous point: it could be important to declare the typology of the hub (hinged or fixed teeter) of the analyzed blades. This could lead to a deeper interpretation of the results. Given the status of the paper, the two-bladed configuration differs from the three-bladed one only for the longer length of the blade. From this point of view, Figure 8 shows a clear trend

in the flutter margins that depends on the sole length of the blade, whereas a different behavior between the two rotor configurations is not visible.

In this work the blade is considered to be rigidly fixed at the root. This is detailed in Section 2, the difference between the two- and three-bladed designs explored here is the inherent structural differences and aerodynamic design variations contributing to the various aero-elastic instabilities. Studying the typology of the hub is important for two-bladed designs, but a full system turbine aero-elastic stability analysis is out of the scope of this work. A continuation of this paper should address the full system aero-elastic-instabilities. The future work in Section 6 has been updated to reflect this.

3. In the redesign section, it should be fair to say that the blades are not redesign with the same constraints of the nominal one. Hence, all design constraints (maximum tip deflection or fatigue) should be verified after the redesign.

Yes, the blades in Section 5 are not designed to the same constraints as the nominal designs. The LE/TE parameter study aims at understanding the flutter trends for the sweep of LE and TE reinforcement layers. The next steps in the process would be to apply appropriate constraints to reduce the number of candidate designs; before proceeding with the final blade design. This has been noted in the last paragraph of Section 5.

Minor comments:
1. Line 5/Abstract: "two-bladed rotors" and "downwind configurations" are both listed as "new rotor concepts". They are not new, but rather "unusual". Please, modify the sentence.

The manuscript has been updated to replace "new rotor concepts" with "non-traditional rotor concepts".

2. Line 111: I think that the name of the flutter computation process is "p-k method" and not "p-f". I could be wrong but, in any case, it is important to indicate the chapter and the section where the method is illustrated in Wright's and Cooper's book.

Thank you, the manuscript has been updated and this typo has been corrected.

3. Line 327: "From these observations of the trends study we note the benefits of additional LE layers to increase the flutter speed versus the TE layers". This is something that could be also foreseen, as a forward motion of the sectional center of gravity is typically beneficial in terms of flap-torsion flutter speed. But my question is: Do Authors think that these results depend on the forward motion of the sectional gravity center or on other structural reasons?

There is a dual effect, one due to the sectional center of mass moving forward, and the general increased stiffness of moving the material from the TE to the LE. And, the impact of these choices on the overall blade mass (which is proportional to blade cost) while examining the trend of flutter margin is shown from this analysis.

4. Line 328: "Similarly, we can have blade designs that are lighter than the baseline structure but have a higher flutter speed". This sentence may be misleading. In fact, the re-design consists in a re-arrangement of the internal blade layout, while other design constraints are not verified (e.g., maximum tip deflection, fatigue etc.…). Hence, it is hard to say that a lighter design is obtain

because the process is performed using the very same constraints of the nominal blade. Please, clarify or remove the sentence.

Yes, the blades in the LE/TE parameters trends study are not designed to the same constraints as the nominal designs. The sentence has been modified to reflect the "trends study" nature of this study, which permits an examination of maximizing flutter speed while minimizing blade mass. Additionally, the last paragraph of Section 5 has been modified to reflect this.

5. Line 377: "it is possible to increase the flutter margins while maintaining or reducing the mass of the blade". Here again, the sentence is clear but prone to misinterpretation as written in the previous comment.

Thank you, this was addressed in the last paragraph of section 5.

6. Conclusions: if relevant, the study of the flutter of the entire turbine could be mentioned within the possible extensions of the present work.

The future work section of this paper has been updated to reflect that a full turbine aero-elastic in-stability analysis is an important extension of this work.